# Local and Adaptive Mirror Descents in Extensive-Form Games

**Côme Fiegel**
CREST - FairPlay, ENSAE Paris
Palaiseau, France
`come.fiege@normalesup.org`

**Pierre Ménard**
ENS Lyon
Lyon, France

**Tadashi Kozuno**
OMRON SINIC X
Tokyo, Japan

**Rémi Munos**
Google DeepMind
Paris, France

**Vianney Perchet**
CREST - FairPlay, ENSAE Paris, Criteo AI Lab
Paris, France

**Michal Valko**
INRIA

## Abstract

We study how to learn $\varepsilon$-optimal strategies in zero-sum imperfect information games (IIG) with *trajectory feedback*. In this setting, players update their policies sequentially, based on their observations over a fixed number of episodes denoted by $T$. As noted by Steinberger et al. (2020) and McAleer et al. (2022), most existing procedures suffer from high variance due to the use of importance sampling over sequences of actions. To reduce this variance, we consider a *fixed sampling* approach, where players still update their policies over time, but with observations obtained through a given fixed sampling policy. Our approach is based on an adaptive Online Mirror Descent (OMD) algorithm that applies OMD locally to each information set, using individually decreasing learning rates and a *regularized loss*. We show that this approach guarantees a convergence rate of $\tilde{\mathcal{O}}(T^{-1/2})$ with high probability and has a near-optimal dependence on the game parameters when applied with the best theoretical choices of learning rates and sampling policies. To achieve these results, we generalize the notion of OMD stabilization, allowing for time-varying regularization with convex increments.

## 1 Introduction

The extensive-form representation of a game (Osborne & Rubinstein, 1994) can be depicted as a tree whose nodes correspond to the game states. At each state, the players choose some available actions and, based on these choices, the game transitions to the next state among the current state's children.

In imperfect information games (IIGs), players may only have access to partial information about the current game state upon taking action. Therefore, the state space is partitioned for each player into multiple information sets, which consist of indistinguishable states from the player's perspective. With perfect recall (Kuhn, 1950), when players remember their previous moves, each space of information sets also has a tree structure.

We focus more specifically on zero-sum IIGs represented in an extensive form under the perfect recall assumption, where the gains of one player, conventionally called the max-player, are equal to the losses of his opponent, the min-player. The primary goal is to design an algorithm learning $\varepsilon$-optimal strategies (von Neumann, 1928). To achieve this, one can use the self-play framework, where an agent controls both players for $T$ episodes. At the beginning of each episode, the agent prescribes a strategy for each player. The agent then observes the play and updates the players' strategies for the next episode based on the outcome of the game. After $T$ episodes, this protocol returns a guess of

38th Conference on Neural Information Processing Systems (NeurIPS 2024).

strategies with a small exploitability gap (Ponsen et al., 2011). In this learning framework, the agent has very limited feedback, only observing the rewards along each sampled trajectory, as opposed to richer feedback that would for example include all possible rewards and all transition probabilities, (Zinkevich et al., 2007; Hoda et al., 2010; Tammelin, 2014; Kroer et al., 2015; Burch et al., 2019) unrealistic in large games.

To deal with this learning framework, a well-studied approach is to unilaterally minimize the regret of each player during the interactions with the game, i.e. the difference between the cumulative gain the player would have obtained had he played the best fixed a posteriori policy and the cumulative gain obtained by following the sequence of policies. The key observation is that by minimizing the regret of both players, the average policies over the sequence of policies generated during the process converge toward optimal strategies at the rate of order $\mathcal{O}(1/\sqrt{T})$ (Cesa-Bianchi & Lugosi, 2006; Kozuno et al., 2021). Regret minimizers such as CFR-based algorithm or online mirror descent (OMD) (Hoda et al., 2010; Kroer et al., 2015) can be used, leading to optimal rates (with respect to the game size) with the latter option (Bai et al., 2022; Fiegel et al., 2023).

Since the agent only observes trajectories of the game, an importance sampling estimate (Auer et al., 2003) of gain (or loss) is fed to the regret minimizer. However, the estimate of this loss usually suffers from high variance due to two reasons. First, the same sequence of policies is used to minimize the regret and to collect the trajectories, making the players strive to fulfill two competing goals: play a policy with small regret and play a policy leading to a small variance gain estimate. Second, importance sampling is applied to sequences of actions, that have in large games a very small probability of being played, leading to empirically large importance sampling weights and ultimately inflating the variance of the gain estimates.

To mitigate this issue, regularization and biasing the estimates can help (Kozuno et al., 2021; Bai et al., 2020). However, the high variance of the gain estimates remains problematic with large games, for which the algorithms are generally coupled with function approximation (Steinberger et al., 2020; McAleer et al., 2022). For instance, neural networks are particularly susceptible to noise (Zhang et al., 2021). A natural question is thus whether it is possible to learn optimal strategies without relying on importance-sampling over the sequence of actions.

To this aim, we consider a particular case of the self-play framework: the fixed policy sampling framework (Lanctot et al., 2009). In this setting, a fixed policy is used to collect the trajectories of the game. Precisely, at each round, one player, let's say the min-player, follows the fixed sampling policy to play against the current policy of the max-player. The collected trajectory is then used to update the current policy of the min-player. In the next episode, the max-player will follow a sampling policy against the current policy of the min-player, and so on. The outcome sampling MCCFR algorithm adopts this framework to update the two players' policy by regret minimization, feeding the CFR algorithm with gain estimated via importance sampling (Lanctot et al., 2009; Bai et al., 2020; Farina et al., 2021b).

Recently, McAleer et al. (2022) proposed the ESCHER algorithm that removes the need for importance sampling in this framework. In particular, as the CFR algorithm is invariant by re-scaling of the gains and the weights of the sampling policy are fixed, ESCHER can directly operate with the unweighted history cumulative gain (Bai et al., 2020). Unfortunately, it still requires access to an oracle that provides this history of cumulative gains at an arbitrary information set.

Nonetheless, the insight of McAleer et al. (2022) cannot be used directly for OMD-based algorithms as they are not scale-invariant. Furthermore, the OMD-based algorithms generally work at the global game level whereas CFR-based algorithms work at the local level of the information set (Bai et al., 2020), making local adaptation to the problem easier.

**Contributions**  We make the following main contributions:

- We propose the `LocalOMD` algorithm, in the fixed policy sampling framework, that allows adaptive learning rates and does not require importance-sampling over the sequence of actions but only for the current action. We explain how it can simply be seen as a regret minimization procedure applied to a local loss on each information set, similarly to Farina et al. (2019b).

- We prove that `LocalOMD` achieves, in this fixed sampling framework, a $\widetilde{\mathcal{O}}\left(1/\varepsilon^2\right)$[1] sample complexity with *any* choice of non-degenerate sampling policy, ignoring the game and policy-dependent parameters.
- With an appropriate sampling policy and choice of learning rates, we prove that `LocalOMD`, recover the $\widetilde{\mathcal{O}}\left(H^3(A_{\mathcal{X}} + B_{\mathcal{Y}})/\varepsilon^2\right)$ near-optimal sample complexity for learning $\varepsilon$-optimal strategies in a tabular setting, where $H$ is the height of the tree, $A_{\mathcal{X}}$ the total number of available actions for the min-player and $B_{\mathcal{Y}}$ the same quantity for the max-player. This sample complexity was also achieved in the fixed policy framework by `BalancedCFR` (Bai et al., 2022), but with a less generalizable procedure that updates the policy at one depth at a time.
- We generalize the dual-stabilization technique introduced by Fang et al. (2020) to analyze OMD with a time-varying regularization as long as the increments of the regularization are convex.
- Our tabular experiments reveal that our algorithm yields comparable results to existing baselines while demonstrating a reduced variance in loss estimation.

## 2 Settings and fixed sampling procedure

### 2.1 Extensive-form games and regret

**Game definition**    We consider a finite zero-sum IIG game $(\mathcal{S}, \mathcal{X}, \mathcal{Y}, \mathcal{A}, \mathcal{B}, p, \ell)$ with perfect recall. Given two behavioral policies $\mu = (\mu(\cdot|x))_{x \in \mathcal{X}}$ and $\nu = (\nu(\cdot|y))_{y \in \mathcal{Y}}$, one episode of such game proceeds as follows: An initial game state $s_1 \sim p(\cdot|s_0)$ is first sampled in the set of states $\mathcal{S}$ according to the transition function $p$, starting from the root $s_0$ of the tree. At depth $h$, the min- and max-players respectively observe the information set $x_h$ and $y_h$ associated with the current state $s_h$ in the spaces of information sets $\mathcal{X}$ and $\mathcal{Y}$ (these spaces being two partitions of $\mathcal{S}$), then simultaneously choose and execute actions $a_h \sim \mu(\cdot|x_h)$ and $b_h \sim \nu(\cdot|y_h)$ in the sets of legal actions $\mathcal{A}(x_h)$ and $\mathcal{B}(y_h)$. As a result, the state transitions to a new state $s_{h+1} \sim p(\cdot|s_h, a_h, b_h)$ in $\mathcal{S}$, with the min- and max- players getting respectively the losses $\ell_h \sim \ell(\cdot|s_h, a_h, b_h)$ in $[0, 1]$ and $1 - \ell_h$ according to the loss distribution $\ell$. This is repeated until a final state $s_H$ of a fixed depth $H$ is reached, after which the episode finishes.

**Policies and actions**    We will denote by $\Pi_{\min}$ and $\Pi_{\max}$ the set of behavioral policies of the min- and max- players. Because of the perfect recall assumption, such policies, with an independent stochastic choice of action for each information set, are enough to describe the entire set of strategies (Laraki et al., 2019). We will also denote by $A_{\mathcal{X}}$ and $B_{\mathcal{Y}}$ the total number of actions for respectively the min- and max- players, i.e. $A_{\mathcal{X}} := \sum_{x \in \mathcal{X}} |\mathcal{A}(x)|$ and $B_{\mathcal{Y}} = \sum_{y \in \mathcal{Y}} |\mathcal{B}(y)|$.

**Regret and $\varepsilon$-optimal strategies**    We are interested in learning $\varepsilon$-optimal policies through self-play over multiple episodes. A useful notion for this objective is the regret as explained in the introduction. We first define the value $V^{\mu,\nu} = \mathbb{E}^{\mu,\nu}[\sum_{h=1}^H \ell_h]$ as the expected sum of losses (for the min-player) with respect to a pair of policies $(\mu, \nu) \in \Pi_{\min} \times \Pi_{\max}$. Given a sequence $(\mu^t, \nu^t)_{t \in [T]}$ in $\Pi_{\min} \times \Pi_{\max}$, the regrets of the min- and max- players are then defined by

$$\mathfrak{R}_{\min}^T := \max_{\mu^\dagger \in \Pi_{\min}} \sum_{t=1}^T (V^{\mu^t, \nu^t} - V^{\mu^\dagger, \nu^t}) \quad \text{and} \quad \mathfrak{R}_{\max}^T := \max_{\nu^\dagger \in \Pi_{\max}} \sum_{t=1}^T (V^{\mu^t, \nu^\dagger} - V^{\mu^t, \nu^t}).$$

Minimizing the regret of both players leads to the computation of an $\varepsilon$-optimal profile (equivalent to an $\varepsilon$-Nash equilibrium for two players zero-sum games) through the computation of an average of the policies. The following theorem quantifies this statement under the perfect recall assumption.

**Theorem 2.1.** *(Cesa-Bianchi & Lugosi, 2006; Kozuno et al., 2021) From a sequence $(\mu^t, \nu^t)_{t \in [T]}$ in $\Pi_{\min} \times \Pi_{\max}$ a certain time-averaged profile $(\overline{\mu}, \overline{\nu})$ is $\varepsilon$-optimal with $\varepsilon = \left(\mathfrak{R}_{\min}^T + \mathfrak{R}_{\max}^T\right)/T$.*

It especially shows that both averaged strategies converge to the set of optimal strategies as long as the regret of both players is sub-linear.

We now focus on the min-player point of view because of the symmetry of the game. Indeed, the following ideas will apply exactly the same way to the max-player, using the losses $1 - \ell_h$ instead.

---

[1]For algorithms with a probability at least $1 - \delta$ of a correct output, the symbol $\widetilde{\mathcal{O}}$ hides dependencies logarithmic in $A_{\mathcal{X}}, B_{\mathcal{Y}}$ and $\delta$

---

**Algorithm 1** Learning procedures with fixed sampling policies for two players

---

1: **Input:** Fixed sampling policies $\mu^s$ and $\nu^s$. Initial policies $\mu^1$ and $\nu^1$ and update procedure for each player
2: For $t = 1$ to $T$
      The min-player observes the full outcome of an episode with the policies $(\mu^s, \nu^t)$
      The max-player observes the full outcome of an episode with the policies $(\mu^t, \nu^s)$
      The min- and max-player respectively update $\mu^{t+1}$ and $\nu^{t+1}$ based on their past observations
3: **Output:** The time-averaged policies $\overline{\mu}, \overline{\nu}$ of Theorem 2.1

---

**Perfect recall and realization plan**  Under the perfect recall assumption, players do not forget their past observations and actions. We can then assume, for any information set $x \in \mathcal{X}$ and action $a \in \mathcal{A}(x)$, the existence of a unique depth $h \in [H]$ and history $(x_1, a_1, ..., x_h, a_h)$ such that $x_h = x$ and $a_h = a$. Using this unique history, we define the realization plan $\mu_{1:} \in \mathbb{R}^{A_\mathcal{X}}$ (von Stengel, 1996) associated to a policy $\mu \in \Pi_{\min}$ with, for any $x \in \mathcal{X}$ and $a \in \mathcal{A}(x)$ by $\mu_{1:}(x,a) := \Pi_{i=1}^h \mu(a_i|x_i)$. It denotes the combined probability of choosing actions that lead to $(x, a)$. We will especially define $Q_{\max} := \{\mu_{1:}, \mu \in \Pi_{\min}\}$ the treeplex, i.e. the set of all possible realization plans.

**Loss and regret linearization**  For $\nu$ a max-player policy, the unique history also allows for the definition of adversarial transitions $p_{1:}^\nu \in \mathbb{R}^\mathcal{X}$ and adversarial losses $\ell^\nu \in \mathbb{R}^{A_\mathcal{X}}$ with:

$$p_{1:}^\nu(x) := p(x_1|s_0) \prod_{i=2}^h p^\nu(x_i|x_{i-1}, a_{i-1}) \quad \text{and} \quad \ell^\nu(x,a) := p_{1:}^\nu(x)\ell_h^\nu(x,a)$$

where $p(x_1|s_0)$ is the probability that $x_1$ is initially observed by the min-player, and, assuming that the max-player policy is set to $\nu$, $p^\nu(\cdot|(x_{i-1}, a_{i-1}))$ denotes the probability of transitioning to $x_i$ when $(x_{i-1}, a_{i-1})$ is reached, and $\ell_h^\nu$ the average loss $\ell_h$ associated to $a$ when $x$ is reached. Similarly to the realization plan, the adversarial transitions denote the combined probability of both Nature and max-player actions that lead to $x$, assuming that the min-player plays the actions $(a_1, ..., a_{h-1})$.

Using a chain-rule argument, we get the relation $V^{\mu,\nu} = \langle \ell^\nu, \mu_{1:} \rangle$, given a pair of policies $(\mu, \nu) \in \Pi_{\min} \times \Pi_{\max}$, where $\langle \cdot, \cdot \rangle$ is the standard inner product of $\mathbb{R}^{A_\mathcal{X}}$, defined by $\langle z_1, z_2 \rangle := \sum_{x \in \mathcal{X}} \sum_{a \in \mathcal{A}(x)} z_1(x,a)z_2(x,a)$. The regret can then be rewritten

$$\mathfrak{R}_{\min}^T = \max_{\mu^\dagger \in \Pi_{\min}} \sum_{t=1}^T \left\langle \ell^t, \mu_{1:}^t - \mu_{1:}^\dagger \right\rangle$$

where $\ell^t := \ell^{\nu^t}$, which effectively reduces the problem to a linear regret problem over the convex polytope $Q_{\min}$ of realization plans.

Several techniques exist to sequentially choose policies $(\mu^t)_{t \in [T]}$ minimizing $\mathfrak{R}_{\min}^T$, assuming that the losses $\ell^t$ are observed after each round $t$ (Hoda et al., 2010). However, in the *trajectory feedback* setting, these losses are not observed, and can only be estimated from the observation of the trajectories $(x_1^t, a_1^t, ..., x_H^t, a_H^t)$ and partial losses $(\ell_1^t, ..., \ell_H^t)$ of each round.

### 2.2  Fixed sampling policy

In the *fixed sampling* framework (Lanctot et al., 2009), both players always use the same policy for the observations of the trajectory. However, the two observations can not be done simultaneously with such an approach, as the learning would then be quite naive. The solution, summarized in Algorithm 1, is for the two players to take turns between an observation phase, in which they play their fixed sampling policy $\mu^s$ or $\nu^s$, and an interaction phase, in which they play their updated policy $\mu^t$ or $\nu^t$. The underlying idea is that the observation phase lets each player observe how the game unfolds against the opponent in its interaction phase, playing its updated policy. Given upper-bounds of the regrets $\mathfrak{R}_{\min}^T$ and $\mathfrak{R}_{\max}^T$ associated to the sequence $(\mu^t, \nu^t)_{t \in [T]}$, the previous Theorem 2.1 then characterizes the optimality of the outputted time-averaged profile $(\overline{\mu}, \overline{\nu})$.

While theoretically optimal algorithms already exist using simultaneous regret minimization procedures (Bai et al., 2022; Fiegel et al., 2023), this framework allows for the removal of the global

importance sampling term of the loss, which reduces the variance to make algorithms more suitable beyond the tabular setting (McAleer et al., 2022). Indeed, as the probability of choosing a sequence of action reaching a given information set is fixed, the average estimations of the losses do not need to be re-weighted based on the inverse of a changing probability. This re-weighting eventually leads to unstable function approximation, e.g. with neural networks, as this probability can be very small.

Furthermore, the fixed sampling framework also allows aggressive policies more focused on exploitation, as the observation side is handled by the sampling strategy. The downside is that this sampling policy must be fixed in advance, which requires defining a good sampling policy beforehand.

From now on, we again focus on the min-player for the same symmetry reasons.

**Estimated regret** Based on the min-player observations, we define $\hat{\mathfrak{R}}_{\min}^T$ the estimated regret by

$$\hat{\mathfrak{R}}_{\min}^T := \max_{\mu^\dagger \in \Pi_{\min}} \sum_{t=1}^T \left\langle \widehat{\ell}^t, \mu_{1:}^t - \mu_{1:}^\dagger \right\rangle$$

where the $\widehat{\ell}^t$ are the importance-sampling estimated loss vectors, defined for each information set $x$ of depth $h$ and action $a \in \mathcal{A}(x)$ by

$$\widehat{\ell}^t(x,a) := \frac{\mathbb{I}_{\left\{x=x_h^t, a=a_h^t\right\}}}{\mu_{1:}^s(x,a)} \ell_h^t$$

with $x_h^t$ the visited information set, $a_h^t$ the chosen action and $\ell_h^t$ the loss at depth $h$ of episode $t$.

The following theorem states that upper-bounding this estimated regret is enough to upper-bound the actual regret, up to an additional additive term. Its proof is given in Appendix B and relies on Bernstein-type inequalities.

**Theorem 2.2.** *Assume that the estimated losses are obtained with a fixed positive sampling policy $\mu^s$ as above. Then, for any sequence $(\mu^t)_{t \in [T]}$ of $\Pi_{\min}$ and any $\delta \in (0,1)$, the following bound holds with a probability at least $1 - \delta$*

$$\mathfrak{R}_{min}^T \le \max\left\{\hat{\mathfrak{R}}_{min}^T, 0\right\} + 4\sqrt{\iota H \kappa(\mu^s) T}$$

*where $\iota := \log\left(\frac{A_\mathcal{X}+1}{\delta}\right)$ and $\kappa(\mu^s) := \max_{\mu \in \Pi_{\min}} \sum_{x \in \mathcal{X}} \sum_{a \in \mathcal{A}_x} \frac{\mu_{1:}(x,a)}{\mu_{1:}^s(x,a)}$.*

A similar proposition is proved by Farina et al. (2020). Our bound is specific to the importance-sampling loss estimator, but tighter by a factor $\sqrt{\kappa(\mu^s)/H}$.

*Remark* 2.3. The quantity $\kappa(\mu^s)$ can be efficiently computed recursively for each of the sub-trees induced by an information set $x \in \mathcal{X}$, and we will denote by $\kappa(\mu^s|x)$ the associated quantities. The same recursion shows that the *balanced policy* $\mu^\star$, which plays proportionally to the total number of actions of each sub-tree, minimizes all these local quantities and satisfies $\kappa(\mu^\star) = A_\mathcal{X}$. The related computations are provided in Appendix C.

## 3 Adaptive Mirror Descent

We shall now focus on the update procedure the min-player can use to minimize this estimated regret. Let us first define some important notions of convex optimization.

**Definition 3.1.** Let $\Omega \subset \mathbb{R}^n$ be a non-empty open convex, and $\overline{\Omega}$ be its closure. A function $\Psi : \overline{\Omega} \to \mathbb{R}$ is said to be Legendre if $\Psi$ is strictly convex, continuously differentiable on $\Omega$ and $\forall y \in \overline{\Omega} \backslash \Omega$, $\lim_{x \to y} \|\nabla \Psi(x)\| = +\infty$ . The Bregman divergence $\mathbf{D}_\Psi : \overline{\Omega} \times \Omega \to \mathbb{R}$ of a Legendre function $\Psi$ is defined as $\mathbf{D}_\Psi(x,y) := \Psi(x) - \Psi(y) - \langle \nabla \Psi(y), x - y \rangle$. The Fenchel conjugate $\Psi^\star : \mathbb{R}^n \to \mathbb{R} \cup \{+\infty\}$ of $\Psi$ is defined by $\Psi^\star(\xi) = \sup_{x \in \overline{\Omega}} \langle \xi, x \rangle - \Psi(x)$.

### 3.1 Online Mirror Descent and dilated entropy

In an extensive-form game with perfect recall, algorithms based on the Online Mirror Descent (OMD) typically compute at each time step $t$ the update

$$\mu^{t+1} = \arg\min_{\mu \in \Pi_{\min}} \left\langle \widehat{\ell}^t, \mu_{1:} \right\rangle + \mathbf{D}_\Psi(\mu_{1:}; \mu_{1:}^t) \tag{OMD}$$

where $\widehat{\ell}^t$ is the estimated loss and $\Psi : Q_{\min} \to \mathbb{R}$ a Legendre regularizer.

**Dilated entropy**   A common choice of such regularizer is the dilated entropy (Hoda et al., 2010; Kroer et al., 2015). It requires for each $x \in \mathcal{X}$ a Legendre regularizer $\Psi_x$ over a convex domain $\overline{\Omega_x} \subset \mathbb{R}^{|\mathcal{A}(x)|}_{\geq 0}$ that contains the simplex $\Delta_{\mathcal{A}(x)} := \left\{ \mu, \sum_{a \in \mathcal{A}(x)} \mu(a) = 1 \right\}$. For a given list of positive weights $\alpha = (\alpha(x))_{x \in \mathcal{X}}$, the dilated entropy $\Psi_\alpha^{\mathrm{dil}}$ satisfies for any $\mu \in \Pi_{\min}$:

$$\Psi_\alpha^{\mathrm{dil}}(\mu_{1:}) := \sum_{x \in \mathcal{X}} \alpha(x) \mu_{1:}(x) \Psi_x \left( \mu(\cdot|x) \right)$$

where $\mu_{1:}(x) := \sum_{a \in \mathcal{A}(x)} \mu_{1:}(x,a)$. Using this dilated entropy as the regularizer, the OMD updates become

$$\mu^{t+1} = \underset{\mu \in \Pi_{\min}}{\arg\min} \left\langle \widehat{\ell}^t, \mu_{1:} \right\rangle + \mathbf{D}_\alpha^{\mathrm{dil}}(\mu_{1:}, \mu_{1:}^t)$$

where $\mathbf{D}_\alpha^{\mathrm{dil}}(\mu_{1:}, \mu_{1:}^t) := \sum_{x \in \mathcal{X}} \alpha(x) \mu_{1:}(x) \mathbf{D}_x(\mu_{1:}(\cdot|x), \mu_{1:}^t(\cdot|x))$ and $(\mathbf{D}_x)_{x \in \mathcal{X}}$ are the individual Bregman divergences of the $(\Psi_x)_{x \in \mathcal{X}}$. The benefits of this regularization are that it efficiently suits the structure of the game and that the associated updates are easily computed recursively, starting from the final states.

## 3.2   Stabilized OMD algorithm

The regularizer $\Psi$ sometimes needs to change over time. For example, when $T$ is unknown, a regularizer of the form $\Psi^t = \Psi/\eta^t$ is usually considered, with $\eta^t = t^{-1/2}$ the learning rate. Fiegel et al. (2023) gives another example of time-varying regularization, adapting the regularization to the game structure that is assumed to be initially unknown. The previous updates (OMD) do not however allow adaptive regularization in general. In fact, even the simple learning rate decrease $\eta^{t+1} = t^{-1/2}$ can lead to a linear regret dependence with time (Orabona & Pál, 2018).

In this part, we shall consider more generally a sequence of Legendre regularizers $(\Psi^t)_{t \in [T]}$ defined on a convex domain $\overline{\Omega} \subset \mathbb{R}^n$, and that the player chooses a sequence of primal iterates $(w^t)_{t \in [T]}$ (respectively the updated realization plans $(\mu_{1:}^t)_{t \in [T]}$ of our settings) in a closed convex set $\mathcal{C}$ (respectively the treeplex $Q_{\min}$) included in $\overline{\Omega}$, according to a sequence of dual increments $(\xi^t)_{t \in [T]}$ in $\mathbb{R}^n$ (respectively the estimated losses $(\widehat{\ell}^t)_{t \in [T]}$) observed sequentially.

Fang et al. (2020) proposed in the presence of non-increasing learning rates, to use a technique called dual-stabilization to recover the classical OMD bounds. We noticed that their updates can be interpreted as

$$w^{t+1} = \underset{w \in \mathcal{C}}{\arg\min} \left\langle \xi^t, w \right\rangle + \mathbf{D}_{\Psi^t}\left(w, w^t\right) + \mathbf{D}_{\Psi^{t+1} - \Psi^t}\left(w, w^1\right) \qquad \text{(GDS-OMD)}$$

with $\Psi^{t+1} - \Psi^t$ incremental functions assumed to be convex, generalizing their special case $\Psi^{t+1} = \Psi/\eta^{t+1}$. The following theorem, proven in Appendix D shows that classical OMD guarantees can be recovered with these updates.

**Theorem 3.2.** *Let $(w^t)_{t \in [T]}$ be a sequence of primal iterates generated by the updates (GDS-OMD), with convex incremental functions. Then for any $w^\dagger \in \overline{\Omega}$,*

$$\sum_{t=1}^T \left\langle \xi^t, w^t - w^\dagger \right\rangle \leq \mathbf{D}_{\Psi^T}(w^\dagger, w^1) + \sum_{t=1}^T \mathbf{D}_{\Psi^{t,\star}}\left(\nabla \Psi^t(w^t) - \xi^t, \nabla \Psi^t(w^t)\right)$$

*where the $(\Psi^{t,\star})_{t \in [T]}$ are the respective Fenchel conjugates of the $(\Psi^t)_{t \in [T]}$.*

Compared to the guarantees obtained with previous adaptive procedures, such as `Ada-MD` (Joulani et al., 2017), the first term of the bound is stated with respect to $w^1$ instead of the sequence $(w^t)_t$, which is important for some $(\Psi^t)_t$ sequences (Orabona & Pál, 2018).

*Remark* 3.3. `AdaGrad` for stochastic gradient descent (Duchi et al., 2011) is an interesting example of regularizatiom with convex increments (and not only through a decreasing learning rate). It uses the adaptive regularization $\Psi^{t+1} = \|\cdot\|^2_{(G^t)^{1/2}}$, where $G^t$ is a positive semi-definite matrix defined with the gradients $g_k$ by either $G^t = \sum_{k=1}^t g_k g_k^T$ or, more efficiently, by $G^t = \mathrm{Diag}\left(\sum_{k=1}^t g_k g_k^T\right)$.

---

**Algorithm 2** `LocalOMD`

---

1: **Input:**
    Sampling policy $\mu^s \in \Pi_{\min}$ and initial policy $\mu^1 \in \Pi_{\min}$
    Bregman divergences $\mathbf{D}_x$ for each information set $x \in \mathcal{X}$
    Sequences of (possibly adaptive) learning rates $(\eta^t(x))_{t,x}$ for each round $t$ and information set $x$.
2: For $t = 1$ to $T$
    Observes the outcome of an episode using the fixed strategy $\mu^s$
    $q_{H+1}^t \leftarrow 0$
    For $h = H$ to 1:
        $\widetilde{\ell}_h^t \leftarrow \mathbb{I}_{\{a=a_h^t\}} \left(\ell_h^t + q_{h+1}^t\right)\Big/\mu^s(a_h^t | x_h^t)$
        $\mu^{t+1}(\cdot|x) \leftarrow \arg\min_{\mu \in \Delta_{\mathcal{A}(x)}} h_x^t(\mu)$
        $q_h^t \leftarrow \min_{\mu \in \Delta_{\mathcal{A}(x)}} h_x^t(\mu)$
    where $h_x^t(\mu) := \left\langle \widetilde{\ell}_h^t, \mu \right\rangle + \frac{1}{\eta^t(x_h^t)}\mathbf{D}_x\left(\mu, \mu^t(\cdot|x_h^t)\right) + \left(\frac{1}{\eta^{t+1}(x_h^t)} - \frac{1}{\eta^{t'+1}(x_h^t)}\right)\mathbf{D}_x\left(\mu, \mu^1(\cdot|x_h^t)\right)$
    and $t'$ is the last round in which $x_h^t$ was visited
    For all non-visited $x \in \mathcal{X}$:
        $\mu^{t+1}(\cdot|x) \leftarrow \mu^t(\cdot|x)$
3: **Output:** The time-averaged policy $\overline{\mu}$

---

**Adaptive dilatation** In the extensive-form game setting based on the dilated entropy $\Psi_\alpha^{\mathrm{dil}}$, this stabilization can be applied to have weights $(\alpha^t(x))_{x \in \mathcal{X}, t \in [T]}$ that vary with times. The convexity assumption of $\Psi_{\alpha^{t+1}}^{\mathrm{dil}} - \Psi_{\alpha^t}^{\mathrm{dil}}$ then rewrites to having locally non-decreasing weights for each $x \in \mathcal{X}$. In this particular case, the updates are obtained with the formula

$$\mu^{t+1} = \arg\min_{\mu \in \Pi_{\min}} \left\langle \widehat{\ell}^t, \mu_{1:} \right\rangle + \mathbf{D}_{\alpha^t}^{\mathrm{dil}}(\mu, \mu^t) + \mathbf{D}_{\alpha^{t+1}-\alpha^t}^{\mathrm{dil}}(\mu, \mu^1). \tag{DDS-OMD}$$

## 4 `LocalOMD` **algorithm**

### 4.1 Algorithm

Let us now consider the fixed sampling framework introduced in Section 2.2. Given a sequence $(\eta^t(x))_{t \in [T]}$ of locally non-increasing learning rates for each $x \in \mathcal{X}$, we introduce the `LocalOMD` algorithm described in Algorithm 2, that uses the updates (DDS-OMD) above with the adaptive weights $\alpha^t(x) = 1/(\mu_{1:}^s(x)\eta^t(x))$. Dividing the loss by the importance sampling term $1/\mu_{1:}^s(x)$ through the learning rates lets it bypass the large variance that this rate can introduce.

**Local loss** This algorithm can be interpreted as one that locally applies the updates (GDS-OMD) using the local loss $\widetilde{\ell}_h^t$, a regularized version of the sum of subsequent losses. Even though this algorithm results from a global minimization procedure, the local loss only uses the probability $\mu^s(a|x)$ of choosing the last action $a \in \mathcal{A}(x)$ in the important sampling, instead of the combined probability $\mu_{1:}^s(x, a)$ of the realization plan. A similar decomposition was observed by Farina et al. (2019a) for the non-stochastic settings, in which both players directly observe the gradient associated with their policies.

For this reason, the local loss will consistently be at most of order $\mathcal{O}(HA)$. Meanwhile, the loss used by Fiegel et al. (2023) can be of order $\mathcal{O}(A_\mathcal{X})$ (approximately $A^H$ in the worst case), even with IX exploration attempting to alleviate the importance sampling issue. This presents a challenge for potential applications involving function approximation, where $A_\mathcal{X}$ becomes very large (McAleer et al., 2022). For instance, such high-variance estimates could lead to highly unstable training dynamics of a policy parametrized with a neural network.

**Complexity** At each iteration, the algorithm only needs to update the policy along the observed trajectory, so the time complexity per iteration is only $H$ times the cost of a local update. If $\mathbf{D}_x$ is the Kullback-Leibler divergence, the local updates then simplify to some Exponential Weights updates

and the total time complexity of an iteration becomes $\mathcal{O}(HA)$, where $A$ is an upper-bound on the local number of actions.

## 4.2 Theoretical analysis

The analysis of `LocalOMD`, detailed in Appendix E is derived from Theorem 3.2 that bounds the estimated regret. The results on the real regret are then obtained with Theorem 2.2. We now present two choices of regularization and their associated guarantees.

**Adaptive rates**  As `LocalOMD` treats each information set $x \in \mathcal{X}$ as a separate problem through the local losses $\widetilde{\ell}_h^t$, an interesting choice is to consider the same adaptive rates that would be used in the $K$-armed bandit problems. The following theorem provides an upper bound in this case.

**Theorem 4.1.** *(Informal, exact statement in Appendix E)*
*For a large class of regularizers $(\Psi_x)_{x \in \mathcal{X}}$ and learning rates $(\eta^t(x))_{x \in \mathcal{X}, t \in [T]}$, the regret has a $\mathcal{O}(\sqrt{T \log(1/\delta)})$ upper bound (hiding the game-dependent terms) with a probability at least $1 - \delta$. Such learning rates include, for all $x \in \mathcal{X}$ of depth $h$,*

$$\eta^t(x) = \eta \left/ \sqrt{\sum_{k=1}^{t} \mathbb{I}_{\left\{x=x_h^k\right\}}} \right. , \quad \text{or the adaptive version } \eta^t(x) = \eta \left/ \sqrt{\sum_{k=1}^{t} \mathbb{I}_{\left\{x=x_h^k\right\}} \left(\widetilde{\ell}_h^k\right)^2} \right. .$$

The adaptive learning rates mentioned for this theorem generally enjoy better performances in practice. Furthermore, they require no initial computation and are easily updated.

**Optimal rates**  The following theorem uses a constant learning rate that locally depends on the $\kappa(\mu^s|x)$ quantities of Remark 2.3, and on the $A := \max_{x \in \mathcal{X}} |\mathcal{A}(x)|$ quantity that upper bounds the local number of available actions on the whole tree.

**Theorem 4.2.** *Using `LocalOMD` with $\mu^1$ as the uniform policy, with the learning rates $\eta^t(x) = \eta/\kappa(\mu^s|x)$ where $\eta = \sqrt{\log(A)\kappa(\mu^s)/(3HT)}$, and with $\Psi_x$ the Shannon entropy $\Psi_x(\mu) = \sum_{a \in \mathcal{A}(x)} \mu(a) \log(\mu(a))$, we have with a probability at least $1 - \delta$ and $\iota = \log(2(A_{\mathcal{X}} + 1)/\delta)$,*

$$\mathfrak{R}_{\min}^T \leq \left(4 + 2\sqrt{3}\right) H^{3/2} \sqrt{\log(A)\iota\kappa(\mu^s)T} .$$

Note that these rates are not adaptive and thus do not require the stabilization introduced in Section 3.2. When using the balanced policy $\mu^\star$ as the sampling policy, for which $\kappa(\mu^\star) = A_{\mathcal{X}}$, we obtain with Theorem 2.1 the rate $\widetilde{\mathcal{O}}\left(H^{3/2}\sqrt{A_{\mathcal{X}}T}\right)$, near-optimal up to the $H$ dependency (Bai et al., 2022).

## 5 Experiments

We implemented `LocalOMD`, with the parameters of Theorem 4.1 and Theorem 4.2, then tested it against the theoretically optimal `BalancedCFR` (Bai et al., 2022) using the balanced policy as the sample policy, and `BalancedFTRL` (Fiegel et al., 2023). The algorithms were compared on three standard benchmark games: Kuhn poker (Kuhn, 1950), Leduc poker (Southey et al., 2005) and liars dice, using the version 1.4 of the OpenSpiel library (Lanctot et al., 2019) under the Apache 2.0 license. The learning rates (and the $IX$ parameters for the relevant algorithms) were optimized independently for each algorithm using a grid search. The code is available at https://github.com/anon5493/LocalOMD-experiments.

The results are given with respect to the total number of episodes used for learning. This technically disadvantages the fixed sampling algorithms, as these require more than one episode at each round $t$ while still performing a single update on the policy of each player. The exploitability gap, along with the variance across the different instances of the simulation, is plotted in Figure 1, top. Note that this variance across the instances is different from the variance of the estimated loss vector $\widehat{\ell}^t$ our method tries to reduce, which is plotted in the Figure 1, down.

Focusing on the exploitability gap, we observe that the two versions of `LocalOMD` behave similarly and constantly beat `BalancedCFR`, mainly because the latter needs to update each depth with independent

samples, thus needing $H$ times more episodes overall. The results of `BalancedFTRL` are more comparable, exhibiting for example better performances on liars dice but worse on Leduc poker.

In the second figure, we observe that the algorithms based on a fixed sampling procedure indeed get a smaller variance in their loss estimation as the sampling policy stays consistently balanced. `BalancedCFR` again gets worse results compared to `LocalOMD` as the losses of each depth are only estimated every $H$ iteration, which increases its variance.

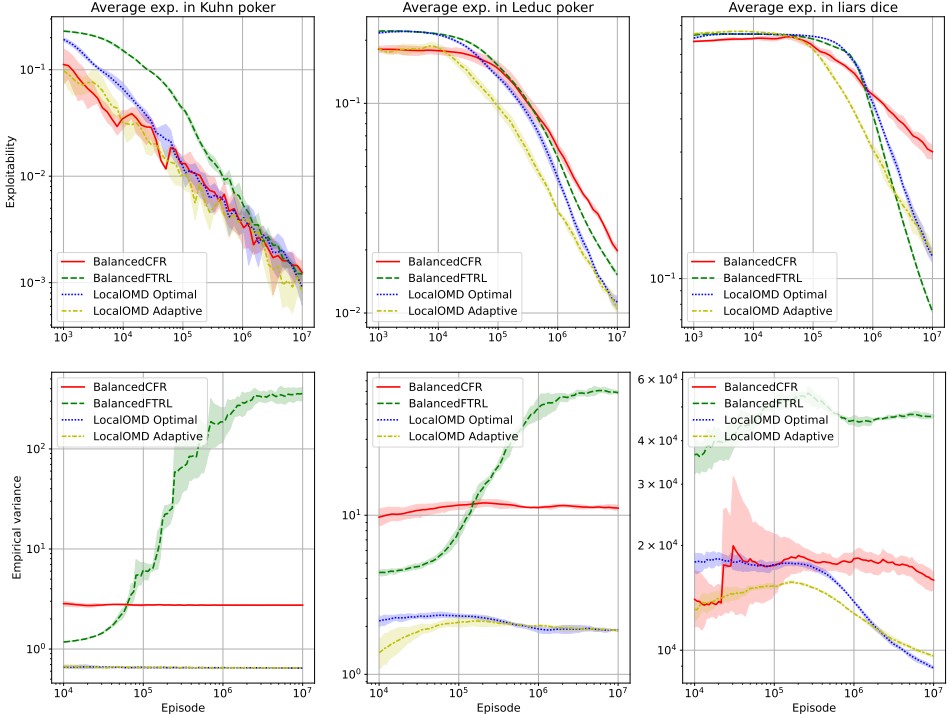

Figure 1: Performances over 5 simulations of various algorithms with respect to the total number of episodes. The vertical axis denotes the exploitability gap $\max_{(\mu,\nu)\in\Pi_{\min}\times\Pi_{\max}} V^{\overline{\mu},\nu} - V^{\mu,\overline{\nu}}$ (top) and the empirical variance of the $\widehat{\ell}^t$ vectors over time (bottom), with all rewards scaled between $0$ and $1$. The total numbers of actions are $A_{\mathcal{X}} = B_{\mathcal{Y}} = 12$ for Kuhn poker, $A_{\mathcal{X}} = B_{\mathcal{Y}} = 1092$ for Leduc poker, and $A_{\mathcal{X}} = B_{\mathcal{Y}} = 24570$ for Liars dice.

# 6 Conclusion

We studied the use of a fixed sampling OMD procedure for the computation of $\varepsilon$-optimal strategies. This approach relies, for each player, on an uncoupling between the observation policy and the interaction policy as described in Algorithm 1. This uncoupling is in direct contrast with the more restrictive semi-bandit setting usually considered for self-play, where these two policies must coincide by design. Notice that this is not the standard exploration/exploitation trade-off, as even in the expert setting (with full information), some kind of exploration is still required.

While the balanced observation policy gets the optimal rates in the worst case, it may not always be the best one for a given game. An alternate choice is to instead use for the observations the current average policy (Gibson et al., 2012). This choice can be adapted to the fixed sampling framework, by restarting the algorithm after a certain number of episodes and using the computed average as the new sampling policy.

The proposed algorithm `LocalOMD` also enjoys simultaneously two interpretations: one as a Mirror Descent type algorithm working at the global level, with a single update performed at each iteration over the whole tree; and one as regret minimizers working locally at each information set, which makes it very similar to a CFR algorithm despite a fundamentally different approach.

We would like to conclude by providing the following interesting research directions.

**Problem-dependent optimality**    For a given game structure and fixed sampling policy $\mu^s$, is there a policy-dependent lower bound $\mathcal{O}(\sqrt{\kappa(\mu^s)T})$ on the regret? We wonder if the $\kappa(\mu^s)$ quantity of Remark 2.3 denotes some sort of complexity related to the problem.

**General sum game**    Using the same techniques as Bai et al. (2022), in a general sum game with potentially more than two players, `LocalOMD` can be shown to converge to an $\varepsilon$-approximate normal-form coarse correlated equilibrium. Are convergences to other forms of correlated equilibrium possible using this fixed sampling policy framework?

**On-policy algorithms**    Is it also possible to remove the importance-sampling of the previous actions in the usual semi-bandit framework that observes with the current policy? The answer is not obvious since the current approach heavily relies on the fact that the sampling policy is fixed.

## 7  Acknowledgements

This research was supported in part by the French National Research Agency (ANR) in the framework of the PEPR IA FOUNDRY project (ANR-23-PEIA-0003) and through the grant DOOM ANR-23-CE23-0002. It was also funded by the European Union (ERC, Ocean, 101071601). Views and opinions expressed are however those of the author(s) only and do not necessarily reflect those of the European Union or the European Research Council Executive Agency. Neither the European Union nor the granting authority can be held responsible for them.

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

# Appendix and Checklist

## A    Related works

In this section, we review previous works on learning an $\varepsilon$-optimal strategy in IIGs.

**Full feedback**    When the game is known, that is the information set structure space, transitions probability, and reward function are provided, a first line of work recasts the setting through the sequence-form representation of a game as a linear program which can be solved efficiently (Romanovsky, 1962; von Stengel, 1996; Koller et al., 1996). A second line of work relies on first-order optimization methods for saddle point computation (Hoda et al., 2010; Kroer et al., 2015, 2018, 2020; Munos et al., 2020; Lee et al., 2021). In particular Hoda et al. (2010); Kroer et al. (2018) relies on the Nesterov smoothing technique Nesterov (2005) whereas Kroer et al. (2015, 2020) use the `MirrorProx` algorithm (Nemirovski, 2004). These methods have a rate of convergence of order $\widetilde{\mathcal{O}}(\text{poly}(H, A_{\mathcal{X}}, B_{\mathcal{Y}})/\varepsilon)$.

A third approach, counterfactual regret minimization (Zinkevich et al., 2007), leverages local regret minimization, i.e. minimizing a type of regret at each information set. Popular algorithms are based on the regret-matching algorithm (Hart & Mas-Colell, 2000; Gordon, 2007) such as `CFR` algorithm (Zinkevich et al., 2007) or based on a close variant of regret-matching, e.g. `CFR+` (Tammelin, 2014; Burch et al., 2019; Farina et al., 2021a). Note that other local regret minimizers could be used, see for example Waugh & Bagnell (2014); Farina et al. (2019b). These algorithms enjoy a guarantee of convergence of order $\widetilde{\mathcal{O}}(\text{poly}(H, A_{\mathcal{X}}, B_{\mathcal{Y}})/\varepsilon^2)$.

Nevertheless, all the methods described above need to explore *the whole information set tree* (or the whole state space) in order to compute one update. The cost of one traversal is of order $\mathcal{O}(X + Y)$ if the transitions and the actions of the other player are sampled; see for example the external-sampling `MCCFR` algorithm (Lanctot et al., 2009).

**Trajectory feedback**    A way to tackle the aforementioned issues is to consider the agnostic setting where the *agent has no prior knowledge of the game and only observes trajectories of the game*. Precisely, the rewards and the transition probabilities are unknown.

**Model-based**    A first method to deal with this limited feedback is to build a *model* of the game and then run any full feedback algorithm in this model. For example, Zhou et al. (2020) use *posterior sampling* (PS, Strens, 2000) to learn a model and then use the `CFR` algorithm in games sampled from the posterior. They obtain a convergence rate of order $\widetilde{\mathcal{O}}(\text{poly}(H, S, A, B)/\varepsilon^2)$ but only when the games are actually sampled according to the known prior. Instead, Zhang & Sandholm (2021) relies on the principle of optimism in the presence of uncertainty to incrementally build a model of the game. Then, the `CFR` algorithm is fed with *optimistic estimates* of the local regrets. They prove a high-probability sample complexity of order $\widetilde{\mathcal{O}}(\text{poly}(H, S, A, B)/\varepsilon^2)$.

**Model-free**    Another line of work (Lanctot et al., 2009; Johanson et al., 2012; Schmid et al., 2018; Farina et al., 2020) directly estimates the local regret via importance sampling that is then fed to the `CFR` algorithm. In particular, the outcome-sampling `MCCFR` (Lanctot et al., 2009; Farina et al., 2020) builds an importance sampling estimate of the counterfactual regret by playing according to a well-chosen *balanced policy*. Intuitively, this policy should ensure to *explore all the information sets*. Note that, depending on the structure of the information set space, playing uniformly over the actions at each information set is not necessarily a good choice. Instead, Farina et al. (2020) propose as a balanced policy to play action with probability proportional to the number of leaves in the sub-tree of possible next information sets. In particular, the outcome-sampling `MCCFR` algorithm requires the knowledge of the information set space structure to build its balanced policy. Nonetheless, in order to obtain $\varepsilon$-optimal strategies with high probability, `MCCFR` needs at most $\widetilde{\mathcal{O}}(H^3(A_{\mathcal{X}} + B_{\mathcal{Y}})/\varepsilon^2)$ realizations of the game (Farina et al., 2020; Bai et al., 2022).

Later, Kozuno et al. (2021) proposed to combine *Online Mirror Descent (`OMD`)* with *dilated Shannon entropy as regularizer* and importance sampling estimate of the losses of a player, see also Farina et al. (2021b). They prove a sample complexity, for the proposed algorithm, `IXOMD`, of order

$\widetilde{\mathcal{O}}(H^2(XA_{\mathcal{X}} + YB_{\mathcal{Y}})/\varepsilon^2)$. Interestingly, they do not need to know in advance the structure of the information set space to obtain this bound. However, the sample complexity of `IXOMD` does not match the lower bound for this setting which is of order $\mathcal{O}((A_{\mathcal{X}} + B_{\mathcal{Y}})/\varepsilon^2)$. Recently, Bai et al. (2022) proposed the `Balanced OMD` algorithm that enjoys also relies on `OMD` but with a dilated entropy weighted by the realization plans of balanced policies as regularizers. For this algorithm, they prove a sample complexity of order $\widetilde{\mathcal{O}}(H^3(A_{\mathcal{X}} + B_{\mathcal{Y}})/\varepsilon^2)$.

**Perfect information Markov game**  Another line of work considers Markov game Kuhn (1953) with *perfect* information and limited feedback. However, it does not assume perfect recall. Sidford et al. (2020); Zhang et al. (2020); Daskalakis et al. (2020); Wei et al. (2021) consider the case where a *generative model* is available whereas Wei et al. (2017); Bai et al. (2020); Xie et al. (2020); Liu et al. (2021) deal with the *trajectory feedback* case. Although this setting is related to ours there is no direct comparison between the two.

## B  Regret estimation

In this section, we aim to establish Theorem 2.2 of the main paper. We start by stating a Bernstein-type inequality that we will use multiple times. It can be found e.g. in Exercise 5.15 by Lattimore & Szepesvári (2020). We provide a short proof below as we did not find any for this precise statement.

**Lemma B.1.** *Let $(U^t)_{t\in[T]}$ be a sequence of random variables with respect to a filtration $\mathcal{F}$, and $\gamma > 0$ be a fixed constant such that for all $t$, $\gamma U^t \leq 1$. Then with a probability of at least $1 - \delta'$:*

$$\sum_{t=1}^{T} \left(U^t - \mathbb{E}\left[U^t\big|\mathcal{F}^{t-1}\right]\right) \leq \gamma \sum_{t=1}^{T} \mathbb{E}\left[(U^t)^2\big|\mathcal{F}^{t-1}\right] + \frac{1}{\gamma}\log(\frac{1}{\delta'})$$

*Proof.* For any $t \in [T]$, using the inequalities $\exp(x) \leq 1 + x + x^2$ for all $x \leq 1$ and $1 + x \leq \exp(x)$ for all $x \in \mathbb{R}$, we have

$$\begin{aligned}
\mathbb{E}\left[\exp\left(\gamma U^t\right)\big|\mathcal{F}^{t-1}\right] &\leq \mathbb{E}\left[1 + \gamma U^t + \gamma^2 (U^t)^2\big|\mathcal{F}^{t-1}\right] \\
&= 1 + \gamma\mathbb{E}\left[U^t\big|\mathcal{F}^{t-1}\right] + \gamma^2\mathbb{E}\left[(U^t)^2\big|\mathcal{F}^{t-1}\right] \\
&\leq \exp\left(\gamma\mathbb{E}\left[U^t\big|\mathcal{F}^{t-1}\right] + \gamma^2\mathbb{E}\left[(U^t)^2\big|\mathcal{F}^{t-1}\right]\right) .
\end{aligned}$$

This implies that the random process $(S_t)_{t\in[T]}$ defined by

$$S_t := \exp\left(\sum_{k=1}^{t} \gamma\left(U^k - \mathbb{E}\left[U^k\big|\mathcal{F}^{k-1}\right]\right) - \sum_{k=1}^{t}\gamma^2\mathbb{E}\left[(U^k)^2\big|\mathcal{F}^{k-1}\right]\right)$$

is a super-martingale, with $S_0 = 1$. Using the Markov inequality, we then get

$$\mathbb{P}\left(\frac{1}{\gamma}\log(S_T) > \frac{1}{\gamma}\log\left(\frac{1}{\delta'}\right)\right) = \mathbb{P}\left(S_T > \frac{1}{\delta'}\right) \leq \delta'\,\mathbb{E}(S_T) \leq \delta'$$

which immediately yields the stated inequality with probability at least $1 - \delta'$. $\qquad\square$

This lemma is then used for Theorem 2.2. The filtration $(\mathcal{F}^t)_{t\in[T]}$ will be used, such that $\mathcal{F}^t$ is the sigma-algebra of all variables of the self-play algorithm up to the execution of episode $t + 1$.

**Theorem B.2.** *Assume that the estimated losses are obtained with a fixed positive sampling policy $\mu^s$ as above. Then, for any sequence $(\mu^t)_{t\in[T]}$ of $\Pi_{\min}$ and any $\delta \in (0,1)$, the following bound holds with a probability at least $1 - \delta$*

$$\mathfrak{R}_{min}^{T} \leq \max\left\{\hat{\mathfrak{R}}_{min}^{T}, 0\right\} + 4\sqrt{\iota H\kappa(\mu^s)T}$$

*where*

$$\iota := \log\left(\frac{A_{\mathcal{X}} + 1}{\delta}\right) \quad \text{and} \quad \kappa(\mu^s) := \max_{\mu\in\Pi_{\min}} \sum_{x\in\mathcal{X}}\sum_{a\in\mathcal{A}_x} \frac{\mu_{1:}(x,a)}{\mu_{1:}^{s}(x,a)} \, .$$

*Proof.* We want to show that, with probability at least $1 - \delta$, that

$$\sum_{t=1}^{t} \left\langle \ell^t - \widehat{\ell}^t, \mu_{1:}^t - \mu_{1:} \right\rangle \leq 4\sqrt{\iota H \kappa(\mu^s) T}$$

holds for all $\mu \in \Pi_{\min}$. Then the property follows after re-organizing the inequality and maximizing over $\mu$. In order to do so, we divide this term into two parts:

$$\sum_{t=1}^{T} \left\langle \ell^t - \widehat{\ell}^t, \mu_{1:}^t - \mu_{1:} \right\rangle = \underbrace{\sum_{t=1}^{T} \left\langle \widehat{\ell}^t - \ell^t, \mu_{1:} \right\rangle}_{\text{EST I}} + \underbrace{\sum_{t=1}^{T} \left\langle \ell^t - \widehat{\ell}^t, \mu_{1:}^t \right\rangle}_{\text{EST II}} .$$

We will furthermore assume that $HT \geq \iota\kappa(\mu^s)$, as otherwise, $4\sqrt{\iota H \kappa(\mu^s) T} \leq 4HT$ and the property immediately follows from $\mathfrak{R}_{\min}^T \leq HT$.

*Upper bound of EST I* For all $x \in \mathcal{X}$ of depth $h$ and $a \in \mathcal{A}(x)$, we apply Lemma B.1 to the random process

$$U_{x,a}^t = \ell_h^t \mathbb{I}_{\left\{x = x_h^t, a = a_h^t\right\}}$$

with $\delta' = \delta/(AX + 1)$ and a fixed $\gamma_1 \in (0, 1]$ we will specify later. This yields, with a probability at least $1 - \delta'$, that

$$\sum_{t=1}^{T} \left( \ell_h^t \mathbb{I}_{\left\{x = x_h^t, a = a_h^t\right\}} - \mathbb{E}\left[ \ell_h^t \mathbb{I}_{\left\{x = x_h^t, a = a_h^t\right\}} \Big| \mathcal{F}^{t-1} \right] \right) \leq \gamma_1 \sum_{t=1}^{T} \mathbb{E}\left[ \left( \ell_h^t \right)^2 \mathbb{I}_{\left\{x = x_h^t, a = a_h^t\right\}} \Big| \mathcal{F}^{t-1} \right] + \frac{\iota}{\gamma_1}$$

$$\leq \gamma_1 \sum_{t=1}^{T} \mathbb{E}\left[ \ell_h^t \mathbb{I}_{\left\{x = x_h^t, a = a_h^t\right\}} \Big| \mathcal{F}^{t-1} \right] + \frac{\iota}{\gamma_1} .$$

By definition of the estimated loss, $\ell_h^t \mathbb{I}_{\left\{x = x_h^t, a = a_h^t\right\}}/\mu_{1:}^s(x, a) = \widehat{\ell}^t(x, a)$. We thus divide by $\mu_{1:}^s(x, a)$ both sides of the inequality, and the unbiasedness of the loss estimator yields

$$\sum_{t=1}^{T} \left[ \widehat{\ell}^t(x, a) - \ell^t(x, a) \right] \leq \gamma_1 \sum_{t=1}^{T} \ell^t(x, a) + \frac{\iota}{\gamma_1 \mu_1^s : (x, a)} .$$

This inequality holds for all $(x, a)$ with a probability of at least $1 - \delta A_{\mathcal{X}}/(A_{\mathcal{X}} + 1)$. Taking the scalar product with any $\mu \in \Pi_{\min}$ then gives

$$\sum_{t=1}^{T} \left\langle \widehat{\ell}^t - \ell^t, \mu_{1:} \right\rangle \leq \gamma_1 \sum_{t=1}^{T} \left\langle \ell^t, \mu_{1:} \right\rangle + \frac{1}{\gamma_1} \sum_{x \in \mathcal{X}} \sum_{a \in \mathcal{A}(x)} \frac{\mu_{1:}(x, a)}{\mu_{1:}^s(x, a)}$$

$$\leq \gamma_1 HT + \frac{\iota}{\gamma_1} \kappa(\mu^s) .$$

Using $\gamma_1 = \sqrt{\iota\kappa(\mu^s)/(HT)} \leq 1$ (by assumption), finally yields

$$\text{EST I} \leq 2\sqrt{\iota H \kappa(\mu^s) T} .$$

*Upper bound of EST II* For this upper bound, we apply Lemma B.1 directly to the sequence $U^t = \left\langle -\widehat{\ell}^t, \mu_{1:}^t \right\rangle$. We now choose $\gamma_2 \in \mathbb{R}_+$ (no further assumption is needed on $\gamma_2$ as the sequence is negative) and apply the lemma to get with probability at least $1 - \delta/(A_{\mathcal{X}} + 1)$

$$\sum_{t=1}^{T}\left\langle \ell^t - \widehat{\ell}^t, \mu_{1:}^t\right\rangle \le \gamma_2 \sum_{t=1}^{T}\mathbb{E}\left[\left\langle \widehat{\ell}^t, \mu_{1:}^t\right\rangle^2 \middle| \mathcal{F}^{t-1}\right] + \frac{\iota}{\gamma_2}$$

$$= \gamma_2 \sum_{t=1}^{T}\mathbb{E}\left[\left(\sum_{h=1}^{H}(\ell_h^t)\sum_{x\in\mathcal{X}}\sum_{a\in\mathcal{A}(x)}\mathbb{I}_{\{x=x_h^t, a=a_h^t\}}\frac{\mu_{1:}^t(x,a)}{\mu_{1:}^s(x,a)}\right)^2 \middle| \mathcal{F}^{t-1}\right] + \frac{\iota}{\gamma_2}$$

$$\text{(Cauchy-Schwarz)} \quad \le \gamma_2 H \sum_{t=1}^{T}\mathbb{E}\left[\sum_{h=1}^{H}(\ell_h^t)^2\sum_{x\in\mathcal{X}}\sum_{a\in\mathcal{A}(x)}\mathbb{I}_{\{x=x_h^t, a=a_h^t\}}\frac{\mu_{1:}^t(x,a)^2}{\mu_{1:}^s(x,a)^2}\middle| \mathcal{F}^{t-1}\right] + \frac{\iota}{\gamma_2}$$

$$\le \gamma_2 H \sum_{t=1}^{T}\mathbb{E}\left[\sum_{h=1}^{H}\ell_h^t\sum_{x\in\mathcal{X}}\sum_{a\in\mathcal{A}(x)}\mathbb{I}_{\{x=x_h^t, a=a_h^t\}}\frac{\mu_{1:}^t(x,a)}{\mu_{1:}^s(x,a)^2}\middle| \mathcal{F}^{t-1}\right] + \frac{\iota}{\gamma_2}$$

$$= \gamma_2 H \sum_{t=1}^{T}\mathbb{E}\left[\sum_{h=1}^{H}\sum_{x\in\mathcal{X}}\sum_{a\in\mathcal{A}(x)}\widehat{\ell}^t(x,a)\frac{\mu_{1:}^t(x,a)}{\mu_{1:}^s(x,a)}\middle| \mathcal{F}^{t-1}\right] + \frac{\iota}{\gamma_2}$$

$$= \gamma_2 H \sum_{t=1}^{T}\sum_{x\in\mathcal{X}}\sum_{a\in\mathcal{A}(x)}\ell^t(x,a)\frac{\mu_{1:}^t(x,a)}{\mu_{1:}^s(x,a)} + \frac{\iota}{\gamma_2}$$

$$\text{(as } \ell^t(x,a)\le 1) \quad \le \gamma_2 H \sum_{t=1}^{T}\sum_{x\in\mathcal{X}}\sum_{a\in\mathcal{A}(x)}\frac{\mu_{1:}^t(x,a)}{\mu_{1:}^s(x,a)} + \frac{\iota}{\gamma_2}$$

$$\le \gamma_2 H \kappa(\mu^s) T + \frac{\iota}{\gamma_2}\,.$$

Taking $\gamma_2 = \sqrt{\frac{\iota}{H\kappa(\mu^s)T}}$ then leads to

$$\sum_{t=1}^{T}\left\langle \ell^t - \widehat{\ell}^t, \mu_{1:}^t\right\rangle \le 2\sqrt{\iota H\kappa(\mu^s)T}\,.$$

Summing the two inequalities yields the inequality of the theorem with a probability of at least $1-\delta$. $\qquad\square$

## C  Balanced policy and $\kappa$

This section deals with the $\kappa(\mu^s)$ and local $\kappa(\mu^s|x)$ of the main paper, and links it to the balanced policy $\mu^\star$.

**Recursive $\kappa$ computation**   Let $\mu^s$ be the positive sample policy. For any $\mu \in \Pi_{\min}$ and $x \in \mathcal{X}$ of depth $h$, we define $\kappa_\mu(\mu^s|x)$ the local sum of ratios against $\mu$ in the subtree induced by $x$, i.e.

$$\kappa_\mu(\mu^s|x) := \sum_{x'\in\mathcal{X}, x \text{ is in the history of } x'}\sum_{a'\in\mathcal{A}(x')}\frac{\mu_{h:}(x',a')}{\mu_{h:}^s(x',a')}$$

where, if $(x_1', a_1'..., x_{h'}', a')$ is the history of $(x', a')$,

$$\mu_{h:}(x',a') := \Pi_{i=h}^{h'}\mu(a_i'|x_i')\,.$$

We then formally define $\kappa(\mu^s|x)$ as $\kappa(\mu^s|x) := \max_{\mu\in\Pi_{\min}}\kappa_\mu(\mu^s|x)$. For any $\mu \in \Pi_{\min}$, the following recursive formula stands

$$\kappa_\mu(\mu^s|x) = \sum_{a\in\mathcal{A}(x)}\frac{\mu(a|x)}{\mu^s(a|x)}\left(1 + \sum_{x'\in\mathcal{X}, x' \text{ directly follows } (x,a)}\kappa_\mu(\mu^s|x')\right)$$

that follows from the definition of $\kappa_\mu(\mu^s|x)$. The same kind of recursion can then be obtained for $\kappa(\mu^s|x)$, because each appearance of $\mu$ in the previous equality can be maximized independently (depending on different information sets). This yields

$$\kappa(\mu^s|x) = \max_{\mu \in \Delta_{\mathcal{A}(x)}} \sum_{a \in \mathcal{A}(x)} \frac{\mu(a)}{\mu^s(a|x)} \left( 1 + \sum_{x' \in \mathcal{X}, x' \text{ directly follows } (x,a)} \kappa(\mu^s|x') \right)$$

$$= \max_{a \in \mathcal{A}(x)} \frac{1}{\mu^s(a|x)} \left( 1 + \sum_{x' \in \mathcal{X}, x' \text{ directly follows } (x,a)} \kappa(\mu^s|x') \right) , \tag{1}$$

which allows for a simple recursive computation of $\kappa(\mu^s|x)$. Finally, once the whole recursive computation is done, $\kappa(\mu^s)$ itself can be computed by, defining $\mathcal{X}_1$ the information sets of depth 1,

$$\kappa(\mu^s) = \sum_{x_1 \in \mathcal{X}_1} \kappa(\mu^s|x_1) .$$

**Balanced policy**   $\kappa(\mu^s|x)$ can also be minimized over $\mu^s \in \Pi_{\min}$ recursively from the leaves using the tree structure. Indeed, for each $x \in \mathcal{X}$, assuming that the minimizers of $\kappa(\mu^s|x')$ are already known for subsequent $x'$, the policy $\mu^s \in \Delta_{\mathcal{A}(x)}$ that minimizes the maximum along the actions $a \in \mathcal{A}(x)$ can be computed from (1). Furthermore, if we define $A^\tau(x,a)$ and $A^\tau(x)$ the total number of actions in the subtrees respectively induced by $(x,a)$ and $x$, i.e.

$$A^\tau(x,a) := 1 + \sum_{x' \in \mathcal{X}, (x,a) \text{ is in the history of } x'} |\mathcal{A}(x')| \quad \text{and} \quad A^\tau(x) := \sum_{a \in \mathcal{A}(x)} A^\tau(x,a) ,$$

we can show that $\min_{\mu^s \in \Pi_{\min}} \kappa(\mu^s|x) = A^\tau(x)$, and that the minimum is attained by the balanced policy $\mu^\star$ defined by

$$\mu^\star(a|x) := \frac{A^\tau(x,a)}{A^\tau(x)} .$$

Indeed, if we assume in (1) that the previous property holds for the $\kappa(\mu^s|x')$, then

$$\kappa(\mu^s|x) = \max_{a \in \mathcal{A}(x)} \frac{1}{\mu^s(a|x)} \left( 1 + \sum_{x' \in \mathcal{X}, x' \text{ directly follows } (x,a)} A^\tau(x') \right) = \max_{a \in \mathcal{A}(x)} \frac{A^\tau(x,a)}{\mu^s(a|x)}$$

and the previous equality is minimized when the $\mu^s(a|x)$ are proportional to the $A^\tau(x,a)$, achieved by the balanced policy $\mu^\star$. With this policy, the same equality gives $\kappa(\mu^\star|x) = A^\tau(x)$, which concludes the induction.

Finally, computing $\kappa(\mu^\star)$ yields

$$\kappa(\mu^\star) = \sum_{x_1 \in \mathcal{X}_1} \kappa(\mu^\star|x_1) = \sum_{x_1 \in \mathcal{X}_1} A^\tau(x_1) = A_{\mathcal{X}} .$$

# D   Generalized dual stabilized online mirror descent

This section will establish the bound related to the updates (GDS-OMD) obtained with any Legendre function.

## D.1   General Bregman divergence properties

We start this section by stating multiple properties of the Bregman divergence $\mathbf{D}_\Psi$ for $\Psi$ a convex function, continuously differentiable on an open $\Omega$ and defined on $\overline{\Omega}$, proved by Cesa-Bianchi & Lugosi (2006).

*Law of cosines :* For any $x \in \overline{\Omega}$ and $w, z \in \Omega$, the following equality holds

$$\mathbf{D}_\Psi(x,w) = \mathbf{D}_\Psi(x,z) + \mathbf{D}_\Psi(z,w) - \langle \nabla\Psi(w) - \nabla\Psi(z), x - z \rangle .$$

---

**Algorithm 3** Generalized dual-stabilized online mirror descent

---

1: **Input:**

A sequence of dual increments $\xi^t$

An open subset $\Omega \in \mathbb{R}^n$ and a closed convex $\mathcal{C}$ of $\overline{\Omega}$

A sequence of Legendre regularizers $(\Psi^t)_{t \in [T]}$ on $\overline{\Omega}$ such that for all $t \in [T]$, $\Psi^{t+1} - \Psi^t$ is convex

An initial primal iterate $w^1 \in \mathcal{C}$

2: **Output:**

A sequence $(w^t)_{t \in [T]}$ of primal iterates

3: **Algorithm:**

For $t = 1$ to $T$

$z^t = \nabla \Psi^t(w^t)$

$y^{t+1} = z^t - \xi^t + \nabla \Psi^{t+1}(w_1) - \nabla \Psi^t(w^1)$

$\hat{w}^{t+1} = \nabla \Psi^{t+1,\star}(y^{t+1})$

$w^{t+1} = \Pi_{\mathcal{C}}^{\Psi^{t+1}}(\hat{w}^{t+1})$

---

*Bregman projection :* For $\mathcal{C}$ a closed convex of $\overline{\Omega}$, and $\Psi$ strictly convex, we can define the Bregman projection $\Pi_{\mathcal{C}}^{\Psi}$ over $\overline{\Omega}$ by

$$\Pi_{\mathcal{C}}^{\Psi}(w) = \arg\min_{z \in \mathcal{C}} \mathbf{D}_{\Psi}(z, w) \,.$$

This Bregman projection satisfies a generalized Pythagorean inequality, for $w \in \Omega$ and $z \in \mathcal{C}$

$$\mathbf{D}_{\Psi}(z, w) \geq \mathbf{D}_{\Psi}(z, \Pi_{\mathcal{C}}^{\Psi}(w)) + \mathbf{D}_{\Psi}(\Pi_{\mathcal{C}}^{\Psi}(w), w)$$

*Fenchel dual :* We defined the Fenchel dual $\Psi^{\star}$ of a Legendre function $\Psi$ for any $\xi \in \mathbb{R}^n$ by

$$\Psi^{\star}(\xi) = \sup_{w \in \overline{\Omega}} \langle \xi, w \rangle - \Psi(w) \,.$$

If we consider $\Omega^{\star} := \nabla\Psi(\Omega)$, it can be shown that $\nabla\Psi^{\star}$ is the inverse function of $\nabla\Psi$ over $\Omega^{\star}$, i.e. for any $w \in \Omega$, $\nabla\Psi^{\star}(\nabla\Psi(w)) = w$. Furthermore, for $w, z \in \Omega$,

$$\mathbf{D}_{\Psi}(w, z) = \mathbf{D}_{\Psi^{\star}}(\nabla\Psi^{\star}(z), \nabla\Psi^{\star}(y)) \,.$$

*Strong convexity:* $\Psi$ is said to be 1-strongly convex with respect to a norm $\|\cdot\|$ if for all $w, z \in \Omega$

$$\Psi(z) \geq \Psi(w) + \langle \nabla\Psi(w), z - w \rangle + \frac{1}{2}\|w - z\|^2 \,.$$

In this case, the Bregman divergence of the Fenchel dual $\Psi^{\star}$ satisfies for any $\xi_1, \xi_2 \in \Omega^{\star}$

$$\mathbf{D}_{\Psi^{\star}}(\xi_1, \xi_2) \leq \|\xi_1 - \xi_2\|_{\star}^2$$

where $\|\cdot\|_{\star}$ is the dual norm of $\|\cdot\|$.

### D.2 GDS-OMD Analysis

We will assume in the following parts that the updates of the following algorithm are properly defined, which happens when all vectors $y^{t+1}$ belong to the Fenchel dual space $\Omega^{t+1,\star} := \nabla\Psi^{t+1}(\Omega)$. We make the same assumption on the regular OMD iterates $z^t - \xi^t$.

We start by giving an equivalent formulation of the updates (GDS-OMD) through Algorithm 3.

**Proposition D.1.** *Algorithm 3 computes the updates* (GDS-OMD) *if they are properly defined, i.e. computes the sequence of primal iterates defined by*

$$w^{t+1} = \arg\min_{w \in \mathcal{C}} \langle \xi^t, w \rangle + \mathbf{D}_{\Psi^t}(w, w^t) + \mathbf{D}_{\Psi^{t+1} - \Psi^t}(w, w^1) \,.$$

*Proof.* By definition of $\hat{w}^{t+1}$ in Algorithm 3, we have for all iterations $t \in [T]$ and $w \in \mathcal{C}$

$$
\begin{aligned}
\mathbf{D}_{\Psi^{t+1}}(w, \hat{w}^{t+1}) &= \Psi^{t+1}(w) - \left\langle \nabla\Psi^{t+1}(\hat{w}^{t+1}), w \right\rangle + C_1 \\
&= \Psi^t(w) + \left(\Psi^{t+1}(w) - \Psi^t(w)\right) - \left\langle y^{t+1}, w \right\rangle + C_1 \\
&= \left\langle \xi^t, w \right\rangle + \left(\Psi^t(w) - \left\langle \nabla\Psi^t(w^t), w \right\rangle\right) + \\
&\qquad \left(\Psi^{t+1}(w) - \Psi^t(w) - \left\langle \nabla\Psi^{t+1}(w^1) - \nabla\Psi^t(w^1), w \right\rangle\right) + C_1 \\
&= \left\langle \xi^t, w \right\rangle + \mathbf{D}_{\Psi^t}\left(w, w^t\right) + \mathbf{D}_{\Psi^{t+1}-\Psi^t}\left(w, w^1\right) + C_2
\end{aligned}
$$

where $C_1$ and $C_2$ are constants independent of the choice of $w$ (but not independent of the other variables). As $w^{t+1} = \arg\min_{w \in \mathcal{C}} \mathbf{D}_{\Psi^{t+1}}(w, \hat{w}^{t+1})$, the updates of Algorithm 3 coincide with the updates (GDS-OMD), as both minimize the same function at each iteration up to an additive constant. $\qquad\square$

The updates of Algorithm 3 are then used to show Theorem 3.2 below. Compared to the ones of McMahan (2017) that also allow adaptive regularization, these updates do not suffer from the potential linear rates observed by Orabona & Pál (2018).

**Theorem D.2.** *Let $(w^t)_{t \in [T]}$ be a sequence of primal iterates generated by the updates* (GDS-OMD), *with convex incremental functions. Then for any $w^\dagger \in \overline{\Omega}$,*

$$
\sum_{t=1}^{T} \left\langle \xi^t, w^t - w^\dagger \right\rangle \leq \mathbf{D}_{\Psi^T}(w^\dagger, w^1) + \sum_{t=1}^{T} \mathbf{D}_{\Psi^{t,\star}}\left(\nabla\Psi^t(w^t) - \xi^t, \nabla\Psi^t(w^t)\right)
$$

*Proof.* We can assume, without any incidence on the $(w^t)_{t \in [T]}$ sequence, that $\Psi^{T+1} = \Psi^T$. We also define for all $t \in [T]$ the notations $\varphi^t = \Psi^{t+1} - \Psi^t$ and

$$
\hat{q}^t = \left\langle \xi^t, \hat{w}^{t+1} \right\rangle + \mathbf{D}_{\Psi^t}(\hat{w}^{t+1}, w^t) + \mathbf{D}_{\varphi^t}(\hat{w}^{t+1}, w^1).
$$

We then divide the sum into a stability and a penalty terms:

$$
\sum_{t=1}^{T} \left\langle \xi^t, w^t - w^\dagger \right\rangle = \underbrace{\sum_{t=1}^{T}\left(\hat{q}^t - \left\langle \xi^t, w^\dagger \right\rangle\right)}_{\text{penalty}} + \underbrace{\sum_{t=1}^{T}\left(\left\langle \xi^t, w^t \right\rangle - \hat{q}^t\right)}_{\text{stability}}
$$

and we look at upper-bounding these two terms.

*Penalty term*: For all $t \in [T]$, using the law of cosines on the Bregman divergences of $\Psi^t$ and $\varphi^t$, we have the two equalities:

$$
\mathbf{D}_{\Psi^t}(w^\dagger, w^t) = \mathbf{D}_{\Psi^t}(w^\dagger, \hat{w}^{t+1}) + \mathbf{D}_{\Psi^t}(\hat{w}^{t+1}, w^t) - \left\langle \nabla\Psi^t(w^t) - \nabla\Psi^t(\hat{w}^{t+1}), w^\dagger - \hat{w}^{t+1} \right\rangle
$$

and

$$
\mathbf{D}_{\varphi^t}(w^\dagger, w^1) = \mathbf{D}_{\varphi^t}(w^\dagger, \hat{w}^{t+1}) + \mathbf{D}_{\varphi^t}(\hat{w}^{t+1}, w^1) - \left\langle \nabla\varphi^t(w^1) - \nabla\varphi^t(\hat{w}^{t+1}), w^\dagger - \hat{w}^{t+1} \right\rangle.
$$

Summing these two equalities, we get

$$
\begin{aligned}
&\mathbf{D}_{\Psi^t}(w^\dagger, w^t) + \mathbf{D}_{\varphi^t}(w^\dagger, w^1) \\
&= \mathbf{D}_{\Psi^{t+1}}(w^\dagger, \hat{w}^{t+1}) + \mathbf{D}_{\Psi^t}(\hat{w}^{t+1}, w^t) + \mathbf{D}_{\varphi^t}(\hat{w}^{t+1}, w^1) - \left\langle \xi^t, w^\dagger - \hat{w}^{t+1} \right\rangle \\
&= \mathbf{D}_{\Psi^{t+1}}(w^\dagger, \hat{w}^{t+1}) + \hat{q}^t - \left\langle \xi^t, w^\dagger \right\rangle
\end{aligned}
$$

as by definition of $\hat{w}^{t+1}$ and $y^{t+1}$,

$$
\nabla\Psi^{t+1}(\hat{w}^{t+1}) = y^{t+1} = -\xi_t + \nabla\Psi^t(w^t) + \nabla\varphi^t(w^1).
$$

Furthermore, as $w^{t+1} = \Pi_{\mathcal{C}}^{t+1}(\hat{w}^{t+1})$, the Pythagorean inequality for the Bregman divergence yields that

$$
\mathbf{D}_{\Psi^{t+1}}(w^\dagger, \hat{w}^{t+1}) \geq \mathbf{D}_{\Psi^{t+1}}(w^\dagger, w^{t+1}) + \mathbf{D}_{\Psi^{t+1}}(w^{t+1}, \hat{w}^{t+1}) \geq \mathbf{D}_{\Psi^{t+1}}(w^\dagger, w^{t+1}).
$$

Injecting this in the previous equality and telescoping leads to

$$\sum_{t=1}^{T} \left(\hat{q}^t - \langle \xi^t, w^\dagger \rangle\right) = \sum_{t=1}^{T} \left(\mathbf{D}_{\Psi^t}(w^\dagger, w^t) + \mathbf{D}_{\varphi^t}(w^\dagger, w^1) - \mathbf{D}_{\Psi^{t+1}}(w^\dagger, \hat{w}^{t+1})\right)$$

$$\leq \sum_{t=1}^{T} \left(\mathbf{D}_{\Psi^t}(w^\dagger, w^t) + \mathbf{D}_{\varphi^t}(w^\dagger, w^1) - \mathbf{D}_{\Psi^{t+1}}(w^\dagger, w^{t+1})\right)$$

$$= \mathbf{D}_{\Psi^{T+1}}(w^\dagger, w^1) - \mathbf{D}_{\Psi^{T+1}}(w^\dagger, w^{t+1})$$

$$\leq \mathbf{D}_{\Psi^T}(w^\dagger, w^1)$$

as $\Psi^T = \Psi^{T+1}$ by definition.

*Stability term*: We first notice, for all $t \in [T]$, that

$$\langle \xi^t, w^t \rangle - \hat{q}^t = \langle \xi^t, w^t - \hat{w}^{t+1} \rangle - \mathbf{D}_{\Psi^t}(\hat{w}^{t+1}, w^t) - \mathbf{D}_{\varphi^t}(\hat{w}^{t+1}, w^1)$$

$$\leq \langle \xi^t, w^t - \hat{w}^{t+1} \rangle - \mathbf{D}_{\Psi^t}(\hat{w}^{t+1}, w^t)$$

$$\leq \langle \xi^t, w^t - \tilde{w}^{t+1} \rangle - \mathbf{D}_{\Psi^t}(\tilde{w}^{t+1}, w^t)$$

where

$$\tilde{w}^{t+1} := \arg\min_{\tilde{w} \in \Omega} \left[\langle \xi^t, \tilde{w} \rangle + \mathbf{D}_{\Psi^t}(\tilde{w}, w^t)\right]$$

is the $\tilde{w}^{t+1}$ iterate that would be obtained using a classical OMD step with $\Psi^t$, without the stabilization. By optimality, it verifies

$$\nabla \Psi^t(\tilde{w}^{t+1}) = \nabla \Psi^t(w^t) - \xi^t$$

and the law of cosines then yields

$$\mathbf{D}_{\Psi^t}(w^t, w^t) = \mathbf{D}_{\Psi^t}(w^t, \tilde{w}^{t+1}) + \mathbf{D}_{\Psi^t}(\tilde{w}^{t+1}, w^t) - \langle \nabla \Psi^t(w^t) - \nabla \Psi^t(\tilde{w}^{t+1}), w^t - \tilde{w}^{t+1} \rangle$$

$$(0) = \mathbf{D}_{\Psi^t}(w^t, \tilde{w}^{t+1}) + \mathbf{D}_{\Psi^t}(\tilde{w}^{t+1}, w^t) - \langle \xi^t, w^t - \tilde{w}^{t+1} \rangle.$$

Plugging this in the first inequality, we directly get

$$\langle \xi^t, w^t \rangle - \hat{q}^t \leq \mathbf{D}_{\Psi^t}(w^t, \tilde{w}^{t+1})$$

and we conclude using

$$\mathbf{D}_{\Psi^t}(w^t, \tilde{w}^{t+1}) = \mathbf{D}_{\Psi^{t,\star}}(\nabla \Psi^t(\tilde{w}^{t+1}), \nabla \Psi^t(w^t))$$

$$= \mathbf{D}_{\Psi^{t,\star}}(\nabla \Psi^t(w^t) - \xi^t, \nabla \Psi^t(w^t)).$$

$\square$

# E  LocalOMD analysis

This section will focus on the dilated entropy approach to extensive-form games, and especially on the updates

$$\mu^{t+1} = \arg\min_{\mu \in \Pi_{\min}} \left\langle \widehat{\ell}^t, \mu_{1:} \right\rangle + \mathbf{D}_{\alpha^t}^{\mathrm{dil}}(\mu, \mu^t) + \mathbf{D}_{\alpha^{t+1}-\alpha^t}^{\mathrm{dil}}(\mu, \mu^1) \qquad \text{(GDS-OMD dilated)}$$

that are used by `LocalOMD`.

## E.1  General analysis

The following proposition shows that each update of this form can be computed recursively starting from the leaves of the tree. It requires for any $t \in [T]$ the vector $q^t$ that satisfies for any $x \in \mathcal{X}$ of depth $h$

$$q^t(x) = \min_{\mu \in \Pi_{\min}} \left\langle \widehat{\ell}^{t,\to x}, \mu_{h:}^{\to x} \right\rangle + \mathbf{D}_{\alpha^t}^{\mathrm{dil},\to\mathrm{x}}(\mu, \mu^t) + \mathbf{D}_{\alpha^{t+1}-\alpha^t}^{\mathrm{dil},\to\mathrm{x}}(\mu, \mu^1)$$

where $\to x$ means that the quantity is considered on the sub-tree induced by $x$ rather than the full information set tree, and $\mu_{h:}$ is defined in Appendix C.

**Proposition E.1.** *Consider the previous updates (*GDS-OMD dilated*) and the vectors $(q^t)_{t\in[T]}$ above. Both $\mu^{t+1}$ and $q^t$ can be computed recursively starting from the leaves of the tree through*

$$\mu^{t+1} = \operatorname*{arg\,min}_{\mu\in\Delta_{\mathcal{A}(x)}} h_x^t(\mu) \quad and \quad q^t(x) = \min_{\mu\in\Delta_{\mathcal{A}(x)}} h_x^t(\mu)$$

*where*

$$h_x^t(\mu) = \left\langle \widetilde{\ell}^t(x,\cdot), \mu \right\rangle + (1/\alpha^t(x))\,\mathbf{D}_x(\mu,\mu^t(\cdot|x)) + \left(1/\alpha^{t+1}(x) - 1/\alpha^t(x)\right)\mathbf{D}_x(\mu,\mu^1(\cdot|x))$$

*and the regularized loss $\widetilde{\ell}^t(x,a)$ is defined by*

$$\widetilde{\ell}^t(x,a) := \widehat{\ell}^t(x,a) + \sum_{x'\in\mathcal{X}\,|\,x'\,directly\,follows\,(x,a)} q^t(x')\,.$$

*Proof.* First, note that $\mu^{t+1}$ is the unique minimizer associated to each $q^t(x_1)$ for $x_1$ the information set of depth $1$. Indeed, each of the sub-tree induced by the $x_1$ can be considered as an independent problem. The idea will be to recursively minimize the $q^t(x)$, starting from the leaves (i.e. the final information sets $x_H$), and compute $\mu^{t+1}(\cdot|x)$ as the associated minimizer at each information set.

This minimization is done through, at each $x \in \mathcal{X}$ of depth $h$, with a decomposition of $q^t(x)$. Indeed, separating the induced tree by $x$ between the root and the rest of the tree leads to

$$q^t(x) = \operatorname*{arg\,min}_{\mu\in\Pi_{\min}} \left\langle \widehat{\ell}^t(x,\cdot), \mu(\cdot|x) \right\rangle + (1/\alpha^t(x))\,\mathbf{D}_x(\mu(\cdot|x),\mu^t(\cdot|x))$$

$$+ \left(1/\alpha^{t+1}(x) - 1/\alpha^t(x)\right)\mathbf{D}_x(\mu(\cdot|x),\mu^1(\cdot|x))$$

$$+ \sum_{a\in\mathcal{A}(x)} \mu(a|x) \sum_{x'\in\mathcal{X}\,|\,x'\,directly\,follows\,(x,a)} \left[ \left\langle \widehat{\ell}^{t,\to x'}, \mu_{h+1:}^{\to x'} \right\rangle + \mathbf{D}_{\alpha^t}^{\mathrm{dil},\to \mathrm{x}'}(\mu,\mu^t) + \mathbf{D}_{\alpha^{t+1}-\alpha^t}^{\mathrm{dil},\to \mathrm{x}'}(\mu,\mu^1) \right]$$

$$= \operatorname*{arg\,min}_{\mu\in\Delta_{\mathcal{A}(x)}} \left\langle \widehat{\ell}^t(x,\cdot), \mu \right\rangle + (1/\alpha^t(x))\,\mathbf{D}_x(\mu,\mu^t(\cdot|x)) + \left(1/\alpha^{t+1}(x) - 1/\alpha^t(x)\right)\mathbf{D}_x(\mu,\mu^1(\cdot|x))$$

$$+ \sum_{a\in\mathcal{A}(x)} \mu(a) \sum_{x'\in\mathcal{X}\,|\,x'\,directly\,follows\,(x,a)} q^t(x')$$

$$= \operatorname*{arg\,min}_{\mu\in\Delta_{\mathcal{A}(x)}} \left\langle \widetilde{\ell}^t(x,\cdot), \mu \right\rangle + (1/\alpha^t(x))\,\mathbf{D}_x(\mu,\mu^t(\cdot|x)) + \left(1/\alpha^{t+1}(x) - 1/\alpha^t(x)\right)\mathbf{D}_x(\mu,\mu^1(\cdot|x))$$

$$= \operatorname*{arg\,min}_{\mu\in\Delta_{\mathcal{A}(x)}} h_x(\mu)$$

as each minimization on $\mu \in \Pi_{\min}$ is done on independent components. This justifies the recursive computation of both $\mu^{t+1}$ and $q^t$. $\qquad\square$

This proposition directly provides the proof of correctness of `LocalOMD`, for which the regularized losses at time step $t$ are non-null only on the trajectory with

$$\widetilde{\ell}^t(x,a) = \frac{\mathbb{I}_{\{x=x_h^t, a=a_h^t\}}}{\mu_{1:}^s(x)} \widetilde{\ell}_h^t\,.$$

We now want to upper(bound the regret associated with this sequence $\mu^t$. The following lemma gives a valuable property that links the regularized loss and the estimated loss.

**Lemma E.2.** *For any policy $\mu' \in \Pi_{\min}$, we have*

$$\left\langle \widetilde{\ell}^t, \mu_{1:}' \right\rangle - \sum_{x\in\mathcal{X}} \mu_{1:}'(x) q^t(x) = \left\langle \widehat{\ell}^t, \mu_{1:}' \right\rangle - \hat{q}^t$$

*where $\hat{q}^t = \min_{\mu\in\Pi_{\min}} \left\langle \widehat{\ell}^t, \mu_{1:} \right\rangle + \mathcal{D}_{\alpha^t}^{\mathrm{dil}}(\mu,\mu^t) + \mathcal{D}_{\alpha^{t+1}-\alpha^t}^{\mathrm{dil}}(\mu,\mu^1)$*

*Proof.* By definition of $\widetilde{\ell}^t$ we have, for any $\mu \in \Pi_{\min}$

$$
\begin{aligned}
\left\langle \widetilde{\ell}^t, \mu'_{1:} \right\rangle &= \left\langle \widehat{\ell}^t, \mu'_{1:} \right\rangle + \sum_{x \in \mathcal{X}} \sum_{a \in \mathcal{A}_x} \mu'_{1:}(x,a) \sum_{x' | (x,a) \to x'} q^t(x') \\
&= \left\langle \widehat{\ell}^t, \mu'_{1:} \right\rangle + \sum_{x \in \mathcal{X}} \sum_{a \in \mathcal{A}_x} \sum_{x' | (x,a) \to x'} \mu'_{1:}(x') q^t(x') \\
&= \left\langle \widehat{\ell}^t, \mu'_{1:} \right\rangle + \sum_{x' \in \mathcal{X} \setminus \mathcal{X}_1} \mu'_{1:}(x') q^t(x') \\
&= \left\langle \widehat{\ell}^t, \mu'_{1:} \right\rangle + \sum_{x' \in \mathcal{X}} \mu'_{1:}(x') q^t(x') - \sum_{x' \in \mathcal{X}_1} q^t(x')
\end{aligned}
$$

in which we identified the components of the second sum as the set of non-initial information sets. We then conclude using $\sum_{x \in \mathcal{X}_1} q^t(x) = \hat{q}^t$ by definition of the $q^t$ terms. $\qquad\square$

This lemma is then used to upper bound the estimated regret of the sequence generated by the updates (GDS-OMD dilated). Indeed, while we could apply Theorem 3.2, the associated stability term, which depends on the Fenchel dual of the dilated entropy, is not easy to upper bound. Nonetheless, the proof of the following theorem is mostly the same but with a slightly different definition of the stability and penalty terms.

**Theorem E.3.** *Let* $(\mu^t)_{t \in [T]}$ *be the sequence of policies generated by the updates* (GDS-OMD dilated). *The following bound holds*

$$
\hat{R}^T \leq \underbrace{\sup_{\mu^\dagger \in \Pi_{\min}} \mathbf{D}_{\alpha^T}^{\mathrm{dil}}(\mu_{1:}^\dagger, \mu_{1:}^1)}_{\text{penalty}} + \underbrace{\sum_{t=1}^T \sum_{x \in \mathcal{X}} \alpha^t(x) \mu_{1:}^t(x) \mathbf{D}_x^\star \left( \nabla \Psi_x(\mu_{1:}^t(\cdot|x)) - \frac{1}{\alpha^t(x)} \widetilde{\ell}^t(x, \cdot), \nabla \Psi_x(\mu_{1:}^t(\cdot|x)) \right)}_{\text{stability}}.
$$

*Proof.* The separation between the stability and the penalty terms is the same as in Theorem 3.2, but with $\hat{q}^t$ (of Lemma E.2) defined after the projection rather than before. This leads to the decomposition

$$
\hat{\mathcal{R}}^T = \underbrace{\max_{\mu^\dagger \in \Pi_{\min}} \sum_{t=1}^T \left( \hat{q}^t - \left\langle \widehat{\ell}^t, \mu_{1:}^\dagger \right\rangle \right)}_{\text{penalty}} + \underbrace{\sum_{t=1}^T \left( \left\langle \widehat{\ell}^t, \mu_{1:}^t \right\rangle - \hat{q}^t \right)}_{\text{stability}}.
$$

*Penalty term:* This part is similar to the general theorem. The optimality of $\mu^{t+1}$ leads to, for any $t \in [T]$,

$$
\nabla \Psi^{t+1}(\mu_{1:}^{t+1}) = -\widehat{\ell}^t - g^t + \nabla \Psi^t(\mu_{1:}^t) + \nabla \varphi^t(\mu_{1:}^1).
$$

where $g^t \in Q_{\max}^\perp$ and $\varphi^t = \Psi^{t+1} - \Psi^t$. We use the same two law of cosines as in Theorem 3.2

$$
\mathbf{D}_{\Psi^t}(\mu_{1:}^\dagger, \mu_{1:}^t) = \mathbf{D}_{\Psi^t}(\mu_{1:}^\dagger, \mu_{1:}^{t+1}) + \mathbf{D}_{\Psi^t}(\mu_{1:}^{t+1}, \mu_{1:}^t) - \left\langle \nabla \Psi^t(\mu_{1:}^t) - \nabla \Psi^t(\mu_{1:}^{t+1}), \mu_{1:}^\dagger - \mu_{1:}^{t+1} \right\rangle
$$

$$
\mathbf{D}_{\varphi^t}(\mu_{1:}^\dagger, \mu_{1:}^1) = \mathbf{D}_{\varphi^t}(\mu_{1:}^\dagger, \mu_{1:}^{t+1}) + \mathbf{D}_{\varphi^t}(\mu_{1:}^{t+1}, \mu_{1:}^1) - \left\langle \nabla \varphi^t(\mu_{1:}^1) - \nabla \varphi^t(\mu_{1:}^{t+1}), \mu_{1:}^\dagger - \mu_{1:}^{t+1} \right\rangle
$$

which yields by summing

$$
\begin{aligned}
\mathbf{D}_{\Psi^t}&(\mu_{1:}^\dagger, \mu_{1:}^t) + \mathbf{D}_{\varphi^t}(\mu_{1:}^\dagger, \mu_{1:}^1) \\
&= \mathbf{D}_{\Psi^{t+1}}(\mu_{1:}^\dagger, \mu_{1:}^{t+1}) + \mathbf{D}_{\Psi^t}(\mu_{1:}^{t+1}, \mu_{1:}^t) + \mathbf{D}_\varphi(\mu_{1:}^{t+1}, \mu_{1:}^1) - \left\langle \widehat{\ell}^t + g^t, \mu_{1:}^\dagger - \mu_{1:}^{t+1} \right\rangle \\
&= \mathbf{D}_{\Psi^{t+1}}(\mu_{1:}^\dagger, \mu_{1:}^{t+1}) + \hat{q}^t - \left\langle \widehat{\ell}^t, \mu_{1:}^\dagger \right\rangle
\end{aligned}
$$

where we used $\left\langle g^t, \mu_{1:}^\dagger - \mu_{1:}^{t+1} \right\rangle = 0$ from the orthogonality. Summing over $t \in [T]$ then gives, by telescoping similarly to the general theorem,

$$\sum_{t=1}^{T} \left( \hat{q}^t - \left\langle \widehat{\ell}^t, \mu_{1:}^\dagger \right\rangle \right) = \sum_{t=1}^{T} \left( \mathbf{D}_{\Psi^t}(\mu_{1:}^\dagger, \mu_{1:}^t) + \mathbf{D}_{\varphi^t}(\mu_{1:}^\dagger, \mu_{1:}^1) - \mathbf{D}_{\Psi^{t+1}}(\mu_{1:}^\dagger, \mu_{1:}^{t+1}) \right)$$

$$= \mathbf{D}_{\Psi^{T+1}}(\mu_{1:}^\dagger, \mu_{1:}^1) - \mathbf{D}_{\Psi^{T+1}}(\mu_{1:}^\dagger, \mu_{1:}^{t+1})$$

$$\leq \mathbf{D}_{\Psi^T}(\mu_{1:}^\dagger, \mu_{1:}^1)$$

*Stability term:* From Lemma E.2 used with $\mu' = \mu^t$, we get an alternative expression of the stability term

$$\left\langle \widehat{\ell}^t, \mu_{1:}^t \right\rangle - \hat{q}^t = \left\langle \widetilde{\ell}^t, \mu_{1:}^t \right\rangle - \sum_{x \in \mathcal{X}} \mu_{1:}^t(x) q^t(x)$$

This shows the stability term can be decomposed in a positive linear combination

$$\left\langle \widehat{\ell}^t, \mu_{1:}^t \right\rangle - \hat{q}^t = \sum_{x \in \mathcal{X}} \mu_{1:}^t(x) \left[ \left\langle \widetilde{\ell}^t(x, \cdot), \mu^t(\cdot | x) \right\rangle - q^t(x) \right]$$

and we will individually upperbound each of the terms of the combination. The method is again similar to the general theorem, but locally with the regularized loss. Defining $\Psi_x^t := \alpha^t(x) \Psi_x$ and $\varphi_x^t := \Psi_x^{t+1} - \Psi_x^t$, we have

$$\left\langle \widetilde{\ell}^t(x, \cdot), \mu^t(\cdot | x) \right\rangle - q^t(x)$$

$$= \left\langle \widetilde{\ell}^t(x, \cdot), \mu^t(\cdot | x) - \mu^{t+1}(\cdot | x) \right\rangle - \mathbf{D}_{\Psi_x^t}(\mu^{t+1}(\cdot | x), \mu^t(\cdot | x)) - \mathbf{D}_{\varphi_x^t}(\mu^{t+1}(\cdot | x), \mu^1(\cdot | x))$$

$$\leq \left\langle \widetilde{\ell}^t(x, \cdot), \mu^t(\cdot | x) - \mu^{t+1}(\cdot | x) \right\rangle - \mathbf{D}_{\Psi_x^t}(\mu^{t+1}(\cdot | x), \mu^t(\cdot | x))$$

$$\leq \left\langle \widetilde{\ell}^t(x, \cdot), \mu^t(\cdot | x) - \tilde{\mu}^{t+1}(\cdot | x) \right\rangle - \mathbf{D}_{\Psi_x^t}(\tilde{\mu}^{t+1}(\cdot | x), \mu^t(\cdot | x))$$

where

$$\tilde{\mu}^{t+1}(\cdot | x) := \arg\min_{\tilde{\mu} \in \Omega_x} \left[ \left\langle \widetilde{\ell}^t(x, \cdot), \tilde{\mu} \right\rangle + \mathbf{D}_{\Psi_x^t}(\tilde{\mu}, \mu^t(\cdot | x)) \right]$$

By optimality, $\tilde{\mu}^{t+1}(\cdot | x)$ verifies

$$\nabla \Psi_x^t(\tilde{\mu}^{t+1}(\cdot | x)) = \nabla \Psi_x^t(\mu^t(\cdot | x)) - \widetilde{\ell}^t(x, \cdot)$$

and the law of cosines yields

$$0 = \mathbf{D}_{\Psi_x^t}(\mu^t(\cdot | x), \mu^t(\cdot | x))$$

$$= \mathbf{D}_{\Psi_x^t}(\mu^t(\cdot | x), \tilde{\mu}^{t+1}(\cdot | x)) + \mathbf{D}_{\Psi_x^t}(\tilde{\mu}^{t+1}(\cdot | x), \mu^t(\cdot | x)) -$$

$$\left\langle \nabla \Psi_x^t(\mu^t(\cdot | x)) - \nabla \Psi_x^t(\tilde{\mu}^{t+1}(\cdot | x)), \mu^t(\cdot | x) - \tilde{\mu}^{t+1}(\cdot | x) \right\rangle$$

$$= \mathbf{D}_{\Psi_x^t}(\mu^t(\cdot | x), \tilde{\mu}^{t+1}(\cdot | x)) + \mathbf{D}_{\Psi_x^t}(\tilde{\mu}^{t+1}(\cdot | x), \mu^t(\cdot | x)) - \left\langle \widetilde{\ell}^t(x, \cdot), \mu^t(\cdot | x) - \tilde{\mu}^{t+1}(\cdot | x) \right\rangle$$

Plugging this in the first inequality, we directly get

$$\left\langle \widetilde{\ell}^t(x, \cdot), \mu^t(\cdot | x) \right\rangle - q^t(x) \leq \mathbf{D}_{\Psi_x^t}(\mu^t(\cdot | x), \tilde{\mu}^{t+1}(\cdot | x))$$

and we get the individual upper bounds with

$$\mathbf{D}_{\Psi_x^t}(\mu^t(\cdot | x), \tilde{\mu}^{t+1}(\cdot | [x)) = \alpha^t(x) \mathbf{D}_{\Psi_x}(\mu^t(\cdot | x), \tilde{\mu}^{t+1}(\cdot | [x))$$

$$= \alpha^t(x) \mathbf{D}_{\Psi_x^\star}(\nabla \Psi_x(\tilde{\mu}^{t+1}(\cdot | [x)), \nabla \Psi_x(\mu^t(\cdot | x)))$$

$$= \alpha^t(x) \mathbf{D}_{\Psi_x^\star} \left( \nabla \Psi_x(\mu^t(\cdot | x)) - \frac{1}{\alpha^t(x)} \widetilde{\ell}^t(x, \cdot), \nabla \Psi_x(\mu^t(\cdot | x)) \right)$$

$\square$

This upper bound on the estimated regret is then used with the learning rates considered in the main article.

## E.2 Optimal rates analysis

We first consider the optimal rates of the main paper.

**Theorem E.4.** *Using* `LocalOMD` *with* $\mu^1$ *as the uniform policy, with the learning rates* $\eta^t(x) = \eta/\kappa(\mu^s|x)$ *where* $\eta = \sqrt{\log(A)\kappa(\mu^s)/(3HT)}$, *and with* $\Psi_x$ *the Shannon entropy* $\Psi_x(\mu) = \sum_{a\in\mathcal{A}(x)} \mu(a) \log(\mu(a))$, *the regret is bounded with a probability at least* $1 - \delta$ *by*

$$\mathfrak{R}_{\min}^T \leq \left(4 + 2\sqrt{3}\right) H^{3/2}\sqrt{\log(A)\iota\kappa(\mu^s)T} \quad \text{where} \quad \iota = \log(2(A_{\mathcal{X}} + 1)/\delta).$$

*Proof.* We apply Theorem E.3, using the relations $\alpha^t(x) = 1/(\mu_{1:}^s(x)\eta^t(x))$ and $\mathbb{I}_{\{x=x_h^t\}}\widetilde{\ell}_h^t = \mu_{1:}^s(x)\widetilde{\ell}^t(x,\cdot)$. We again separately bound the penalty and stability terms.

*Penalty term :* We will denote by PEN this term defined by

$$\text{PEN} := \sup_{\mu^\dagger \in \Pi_{\min}} \mathbf{D}_{\alpha^T}^{\text{dil}}(\mu_{1:}^\dagger, \mu_{1:}^1).$$

By definition of the dilated entropy, we have, using that $\mu^1$ is the uniform policy and that the Bregman divergence of the Shannon entropy is the Kullback-Leibler divergence,

$$\text{PEN} = \sup_{\mu^\dagger \in \Pi_{\min}} \sum_{x\in\mathcal{X}} \frac{\mu_{1:}^\dagger(x)\kappa(\mu^s|x)}{\eta\mu_{1:}^s(x)} \mathbf{D}_\Psi(\mu^\dagger(\cdot|x), \mu^1(\cdot|x))$$

$$= \frac{1}{\eta} \sup_{\mu^\dagger \in \Pi_{\min}} \sum_{x\in\mathcal{X}} \frac{\mu_{1:}^\dagger(x)\kappa(\mu^s|x)}{\mu_{1:}^s(x)} \sum_{a\in\mathcal{A}(x)} \mu^\dagger(a|x) \log(\mu^\dagger(a|x)/\mu^1(a|x))$$

$$\leq \frac{1}{\eta} \sup_{\mu^\dagger \in \Pi_{\min}} \sum_{x\in\mathcal{X}} \frac{\mu_{1:}^\dagger(x)\kappa(\mu^s|x)}{\mu_{1:}^s(x)} \sum_{a\in\mathcal{A}(x)} \mu^\dagger(a|x) \log(1/\mu^1(a|x))$$

$$\leq \frac{\log(A)}{\eta} \sup_{\mu^\dagger \in \Pi_{\min}} \sum_{x\in\mathcal{X}} \frac{\mu_{1:}^\dagger(x)}{\mu_{1:}^s(x)} \kappa(\mu^s|x)$$

$$= \frac{\log(A)}{\eta} \sup_{\mu^\dagger \in \Pi_{\min}} \sum_{h=1}^{H} \sum_{x\in\mathcal{X}_h} \frac{\mu_{1:}^\dagger(x)}{\mu_{1:}^s(x)} \sup_{\mu' \in \Pi_{\min}} \sum_{x'\in\mathcal{X}|x \text{ is in the history of } x'} \sum_{a'\in\mathcal{A}(x')} \frac{\mu_{h:}'(x', a')}{\mu_{h:}^s(x', a')}$$

$$\leq \frac{\log(A)}{\eta} \sum_{h=1}^{H} \sup_{\mu^\dagger \in \Pi_{\min}} \sum_{x\in\mathcal{X}_h} \frac{\mu_{1:}^\dagger(x)}{\mu_{1:}^s(x)} \sup_{\mu' \in \Pi_{\min}} \sum_{x'\in\mathcal{X}|x \text{ is in the history of } x'} \sum_{a'\in\mathcal{A}(x')} \frac{\mu_{h:}'(x', a')}{\mu_{h:}^s(x', a')}$$

$$\text{(by independance)} = \frac{\log(A)}{\eta} \sum_{h=1}^{H} \sup_{\mu^\dagger \in \Pi_{\min}} \sum_{x\in\mathcal{X}_h} \frac{\mu_{1:}^\dagger(x)}{\mu_{1:}^s(x)} \sum_{x'\in\mathcal{X}|x \text{ is in the history of } x'} \sum_{a'\in\mathcal{A}(x')} \frac{\mu_{h:}^\dagger(x', a')}{\mu_{h:}^s(x', a')}$$

$$= \frac{\log(A)}{\eta} \sum_{h=1}^{H} \sup_{\mu^\dagger \in \Pi_{\min}} \sum_{x\in\mathcal{X}_h} \sum_{x'\in\mathcal{X}|x \text{ is in the history of } x'} \sum_{a'\in\mathcal{A}(x')} \frac{\mu_{1:}^\dagger(x', a')}{\mu_{1:}^s(x', a')}$$

$$= \frac{\log(A)}{\eta} \sum_{h=1}^{H} \sup_{\mu^\dagger \in \Pi_{\min}} \sum_{x'\in\mathcal{X}} \sum_{a'\in\mathcal{A}(x')} \frac{\mu_{1:}^\dagger(x', a')}{\mu_{1:}^s(x', a')}$$

$$= \frac{\log(A)}{\eta} H\kappa(\mu^s)$$

where $\mathcal{X}_h$ is the set of information sets of depth $h$, the two sums being later merged on the basis of perfect recall. We now look at the stability term.

*Stability term :* We will denote by STA this term defined by

$$\text{STA} := \sum_{t=1}^{T} \sum_{x\in\mathcal{X}} \alpha^t(x)\mu_{1:}^t(x)\mathbf{D}_x^\star\left(\nabla\Psi_x(\mu_{1:}^t(\cdot|x)) - \frac{1}{\alpha^t(x)}\widetilde{\ell}^t(x,\cdot), \nabla\Psi_x(\mu_{1:}^t(\cdot|x))\right)$$

We first look at an upper-bound of $\mathbf{D}_x^\star\left(\nabla\Psi_x(\mu_{1:}^t(\cdot|x)) - \frac{1}{\alpha^t(x)}\widetilde{\ell}^t(x,\cdot), \nabla\Psi_x(\mu_{1:}^t(\cdot|x))\right)$. In order to do so, we upper-bound (in the symmetric matrix sense) the Hessian of $\Psi_x^\star$ on $I :=$ $\left\{\nabla\Psi_x(\mu_{1:}^t(\cdot|x)) - \frac{\gamma}{\alpha^t(x)}\widetilde{\ell}^t(x,\cdot)\Big|\gamma\in[0,1]\right\}$.

Because $\Psi_x(\mu) = \sum_{a\in\mathcal{A}(x)}\mu(a)\log(\mu(a))$ is the Shannon entropy,

$$\nabla\Psi_x(\mu)(a) = \log(\mu(a)) + 1 \quad\text{and thus}\quad \nabla^2\Psi_x(\mu) = \mathrm{Diag}\{(1/\mu(a))\}_{a\in\mathcal{A}(x)}$$

and the Hessian of $\Psi_x^\star$ is given by

$$\nabla^2\Psi^\star(y) = \nabla^2\Psi_x(y)^{-1} = \mathrm{Diag}\{y(a)\}_{a\in\mathcal{A}(x)}\,.$$

In particular, it is upper bounded on $I$ by the matrix $D_\mu$ defined by

$$D_\mu := \mathrm{Diag}\{\mu(a)\}_{a\in\mathcal{A}(x)}$$

This yields that

$$\mathbf{D}_x^\star\left(\nabla\Psi_x(\mu_{1:}^t(\cdot|x)) - \frac{1}{\alpha^t(x)}\widetilde{\ell}^t(x,\cdot), \nabla\Psi_x(\mu_{1:}^t(\cdot|x))\right) \leq \frac{1}{2}\left\|\frac{1}{\alpha^t(x)}\widetilde{\ell}^t(x,\cdot)\right\|_{D_\mu^t(\cdot|x)}^2$$

$$= \frac{1}{2\alpha^t(x)^2}\sum_{a\in\mathcal{A}(x)}\mu^t(a|x)\widetilde{\ell}^t(x,a)^2$$

which leads to

$$\mathrm{STA} \leq \sum_{t=1}^T\sum_{x\in\mathcal{X}}\frac{\mu_{1:}^t(x)}{2\alpha^t(x)}\sum_{a\in\mathcal{A}(x)}\mu^t(a|x)\widetilde{\ell}^t(x,a)^2$$

$$= \frac{\eta}{2}\sum_{t=1}^T\sum_{x\in\mathcal{X}}\mathbb{I}_{\left\{x=x_h^t\right\}}\frac{\mu_{1:}^t(x)}{\mu_{1:}^s(x)}\frac{1}{\kappa(\mu^s|x)}\sum_{a\in\mathcal{A}(x)}\mathbb{I}_{\left\{a=a_h^t\right\}}\mu^t(a|x)\widetilde{\ell}_h^t(a)^2\,.$$

We can first notice from recursively comparing the minimizer $\mu^{t+1}(\cdot|x_h^t)$ with $\mu^t(\cdot|x_h^t)$ that the regularized loss $\widetilde{\ell}_h^t(a_h^t)$, satisfies

$$\widetilde{\ell}_h^t(a_h^t) \leq \left\langle\widehat{\ell}^{t,\to x}, \mu_{h+1:}^{t,\to x}\right\rangle\,,$$

re-using the notation at the beginning of the section, because the regularization does not evolve with time. The difficulty is now to upper bound STA with high probability. In order to do so, we use the Lemma B.1 on the sequence $(U^t)_{t\in[T]}$ defined by

$$U^t := \sum_{x\in\mathcal{X}}\mathbb{I}_{\left\{x=x_h^t\right\}}\frac{\mu_{1:}^t(x)}{\mu_{1:}^s(x)}\frac{1}{\kappa(\mu^s|x)}\sum_{a\in\mathcal{A}(x)}\mathbb{I}_{\left\{a=a_h^t\right\}}\mu^t(a|x)\widetilde{\ell}_h^t(a)^2$$

with $\gamma' = \gamma \in (0, 1/(H^2\kappa(\mu^s))]$ and $\delta' = \delta/2$. This yields with probability at least $1-\delta/2$

$$\sum_{t=1}^T U^t \leq \sum_{t=1}^T\mathbb{E}\left[U^t\big|\mathcal{F}^{t-1}\right] + \gamma\sum_{t=1}^T\mathbb{E}\left[(U^t)^2\big|\mathcal{F}^{t-1}\right] + \iota/\gamma\,.$$

On the one hand, we have, using $\widehat{\ell}_h^t(a_h^t) \leq \kappa(\mu^s|x)$ and the previous inequality that

$$\mathbb{E}\left[U^t\big|\mathcal{F}^{t-1}\right] \leq \sum_{x\in\mathcal{X}}p^t(x)\mu^t(x)\sum_{a\in\mathcal{A}(x)}\left\langle\ell^{t,\to x}, \mu_{h:}^{t,\to x}\right\rangle$$

$$\leq \sum_{x\in\mathcal{X}}p^t(x)\mu^t(x)H$$

$$\leq H^2\,.$$

On the other hand, using the same inequality,

$$
\begin{aligned}
\mathbb{E}\left[(U^t)^2\big|\mathcal{F}^{t-1}\right] &= \mathbb{E}\left[\left(\sum_{x\in\mathcal{X}}\mathbb{I}_{\{x=x_h^t\}}\frac{\mu_{1:}^t(x)}{\mu_{1:}^s(x)}\sum_{a\in\mathcal{A}(x)}\mathbb{I}_{\{a=a_h^t\}}\left\langle\widehat{\ell}_h^t(a),\mu^t(a|x)\right\rangle\right)^2\Bigg|\mathcal{F}^{t-1}\right] \\
&\leq H\mathbb{E}\left[\sum_{x\in\mathcal{X}}\mathbb{I}_{\{x=x_h^t\}}\frac{\mu_{1:}^t(x)^2}{\mu_{1:}^s(x)^2}\sum_{a\in\mathcal{A}(x)}\mathbb{I}_{\{a=a_h^t\}}\left\langle\widehat{\ell}_h^t(a)^2,\mu^t(a|x)^2\right\rangle\Bigg|\mathcal{F}^{t-1}\right] \\
&\leq H\kappa(\mu^s)\mathbb{E}\left[\sum_{x\in\mathcal{X}}\mathbb{I}_{\{x=x_h^t\}}\frac{\mu_{1:}^t(x)}{\mu_{1:}^s(x)}\sum_{a\in\mathcal{A}(x)}\mathbb{I}_{\{a=a_h^t\}}\left\langle\widehat{\ell}_h^t(a),\mu^t(a|x)\right\rangle\Bigg|\mathcal{F}^{t-1}\right] \\
&\leq H\kappa(\mu^s)\sum_{x\in\mathcal{X}}p^t(x)\mu^t(x)\sum_{a\in\mathcal{A}(x)}\left\langle\ell^{t,\to x},\mu_{h:}^{t,\to x}\right\rangle \\
&\leq H\kappa(\mu^s)\sum_{x\in\mathcal{X}}p^t(x)\mu^t(x)H \\
&\leq H^3\kappa(\mu^s)\,.
\end{aligned}
$$

The following upper bound on the stability term thus holds

$$
\text{STA} \leq \eta\left(H^2T+\gamma H^3\kappa(\mu^s)T+\frac{\iota}{\gamma}\right)\,.
$$

Taking $\gamma = 1/(H^2\kappa(\mu^s))$, we obtain

$$
\text{STA} \leq \eta\left(2H^2T+H^2\iota\kappa(\mu^s)\right)
$$

As the bound of the theorem trivially holds if $T < \iota\kappa(\mu^s)$ (the regret being bounded by $T$ anyway), we even have assuming $T \geq \iota\kappa(\mu^s)$

$$
\text{STA} \leq 3\eta H^2T\,.
$$

*Conclusion:* Combining all the previous bounds, the estimated regret is bounded, with a probability of at least $1 - \delta/2$ by

$$
\hat{\mathfrak{R}}^T \leq \frac{\log(A)}{\eta}H\kappa(\mu^s)+3\eta H^2T\,.
$$

Taking $\eta = \sqrt{\log(A)\kappa(\mu^s)/(3HT)}$, we obtain

$$
\hat{\mathfrak{R}}^T \leq 2\sqrt{3}\,H^{3/2}\sqrt{\log(A)\iota\kappa(\mu^s)T}\,.
$$

We finally conclude by combining this bound with Theorem 2.2 for the true regret, using $\delta' = \delta/2$, such that the two inequalities hold with a probability at least $1 - \delta$. $\qquad\square$

### E.3 Adaptive rates analysis

We end this appendix by considering the adaptive setting. We will assume that all regularizers $\Psi_x$ are 1-strongly convex with respect to some norms $\|\cdot\|_x$, and we will define

$$
\begin{aligned}
C_\Psi &:= \sup_{x\in\mathcal{X},\mu\in\Delta_{A_x}}\mathbf{D}_x(\mu,\mu^1(\cdot|x)) \\
C_\Psi^\star &:= \sup_{x\in\mathcal{X},a\in\mathcal{A}_x}\|\mathbb{I}_{\{x,a\}}\|_x^\star
\end{aligned}
$$

where $\mu^1$ is the initial policy considered in the algorithm and $\mathbb{I}_{\{x,a\}}$ is the loss vector $\ell(x,\cdot)$ equal to 1 for $a \in \mathcal{A}(x)$ and 0 for $a' \in \mathcal{A}(x)\backslash\{a\}$. The following theorem is the formal statement of Theorem 4.1 in the main article. While being quite general, the upper bound is unsurprisingly not as tight as the previous one.

**Theorem E.5.** *With such regularizers, assume that the learning rates are locally decreasing and let $\lambda_1, \lambda_2 \in \mathbb{R}_{>0}$ be two constants such that for all information set $x \in \mathcal{X}$,*

$$\max_{t \in [T-1]} \left[ \frac{1}{\eta^{t+1}(x)} - \frac{1}{\eta^t(x)} \right] \leq \lambda_1 \quad \text{and} \quad 1/\eta^T(x) + \sum_{t=1}^T \eta^t(x) \mathbb{I}_{\{x=x_h^t\}} \leq \lambda_2 \sqrt{T}$$

*Then with a probability at least $1 - \delta$, the regret of Algorithm 2 is upper-bounded by*

$$\mathfrak{R}_{\max}^T \leq \left[ 2 \left[ (1 + \lambda_1) C_\Psi C_\Psi^\star \kappa(\mu^s) \right]^2 \lambda_2 |\mathcal{X}| + 4\sqrt{H\kappa(\mu^s)\iota} \right] \sqrt{T}$$

*where $\iota = \log((A_\mathcal{X} + 1)/\delta)$.*

The proof of this theorem will be based on the following lemma that bounds the regularized loss using the $\lambda_1$ constant above.

**Lemma E.6.** *For all $t \in [T]$ and $h \in [H]$,*

$$\widetilde{\ell}_h^t(a_h^t) \leq (1 + \lambda_1 C_\Psi)\kappa(\mu^s|x_h^t).$$

*Proof.* The proof is done recursively on $h$, starting from the leaves. Indeed, for $h = H$, the property is immediate as $\widetilde{\ell}_h^t(a_H^t) \leq 1/\mu^s(a_H^t|x_H^t) \leq \kappa(\mu^s|x_H^t)$. If we assume that the property holds for a depth $h > 1$, then

$$
\begin{aligned}
q_h^t &= \min_{\mu \in \Delta_{\mathcal{A}(x_h^t)}} \left\langle \widetilde{\ell}_h^t, \mu \right\rangle + \frac{1}{\eta^t(x_h^t)} \mathbf{D}_x \left( \mu, \mu^t(\cdot|x_h^t) \right) + \left( \frac{1}{\eta^{t+1}(x_h^t)} - \frac{1}{\eta^t(x_h^t)} \right) \mathbf{D}_x \left( \mu, \mu^1(\cdot|x_h^t) \right) \\
&\leq \left\langle \widetilde{\ell}_h^t, \mu^t(\cdot|x_h^t) \right\rangle + \left( \frac{1}{\eta^{t+1}(x_h^t)} - \frac{1}{\eta^t(x_h^t)} \right) \mathbf{D}_x \left( \mu^t(\cdot|x_h^t), \mu^1(\cdot|x_h^t) \right) \\
&\leq \widetilde{\ell}_h^t(a_h^t) + \lambda_1 C_\Psi.
\end{aligned}
$$

Then

$$
\begin{aligned}
\widetilde{\ell}_{h-1}^t(a_{h-1}^t) &= (\ell_{h-1}^t + q_h^t)/\mu^s(a_{h-1}^t|x_{h-1}^t) \\
&\leq (1 + \lambda_1 C_\Psi + \widetilde{\ell}_h^t(a_h^t))/\mu^s(a_{h-1}^t|x_{h-1}^t) \\
&\leq (1 + \lambda_1 C_\Psi)(1 + \kappa(\mu^s|x_h^t))/\mu^s(a_{h-1}^t|x_{h-1}^t) \\
&\leq (1 + \lambda_1 C_\Psi)\kappa(\mu^s|x_{h-1}^t)
\end{aligned}
$$

which concludes the induction.

$\square$

*Proof.* We now prove the theorem. We start with the estimated regret, that we decompose between the penalty term and the stability term using theorem E.3.

*Penalty term:* The penalty term PEN is bounded by

$$
\begin{aligned}
\text{PEN} &\leq \sup_{\mu^\dagger \in \Pi_{\min}} \mathbf{D}_{\alpha^T}^{\text{dil}}(\mu_{1:}^\dagger, \mu_{1:}^1) \\
&\leq \sup_{\mu^\dagger \in \Pi_{\min}} \sum_{x \in \mathcal{X}} \frac{1}{\eta^T(x)} \frac{\mu_{1:}^\dagger(x)}{\mu_{1:}^s(x)} \mathbf{D}_x(\mu^\dagger(\cdot|x), \mu^1(\cdot|x)) \\
&\leq C_\Psi \lambda_2 \sqrt{T} \sup_{\mu^\dagger \in \Pi_{\min}} \sum_{x \in \mathcal{X}} \frac{\mu_{1:}^\dagger(x)}{\mu_{1:}^s(x)} \\
&\leq C_\Psi \lambda_2 \kappa(\mu^s)\sqrt{T}.
\end{aligned}
$$

*Stability term:* For the stability term STA, we rely on Lemma E.6 and the 1-strong convexity of $\Psi_x$ with respect to $\|\cdot\|_x$ (see Appendix D.1) and get

$$\text{STA} = \sum_{t=1}^{T} \sum_{x \in \mathcal{X}} \alpha^t(x) \mu_{1:}^t(x) \mathbf{D}_x^\star \left( \nabla \Psi_x(\mu_{1:}^t(\cdot|x)) - \frac{1}{\alpha^t(x)} \widetilde{\ell}^t(x, \cdot), \nabla \Psi_x(\mu_{1:}^t(\cdot|x)) \right)$$

$$\leq \sum_{t=1}^{T} \sum_{x \in \mathcal{X}} \frac{\mu_{1:}^t(x)}{\alpha^t(x)} \|\widetilde{\ell}^t(x, \cdot)\|_x^{\star 2}$$

$$\leq \sum_{t=1}^{T} \sum_{x \in \mathcal{X}} \eta^t(x) \mathbb{I}_{\left\{x = x_h^t\right\}} \frac{\mu_{1:}^t(x)}{\mu_{1:}^s(x)} \|\widetilde{\ell}_h^t\|_x^{\star 2}$$

$$\leq [C_\Psi^\star]^2 \sum_{t=1}^{T} \sum_{x \in \mathcal{X}} \eta^t(x) \mathbb{I}_{\left\{x = x_h^t\right\}} \frac{\mu_{1:}^t(x)}{\mu_{1:}^s(x)} \left( \widetilde{\ell}_h^t(a_h^t) \right)^2$$

$$\leq [(1 + \lambda_1) C_\Psi C_\Psi^\star]^2 \sum_{t=1}^{T} \sum_{x \in \mathcal{X}} \eta^t(x) \mathbb{I}_{\left\{x = x_h^t\right\}} \frac{\mu_{1:}^t(x)}{\mu_{1:}^s(x)} \kappa(\mu^s|x)^2$$

$$\leq [(1 + \lambda_1) C_\Psi C_\Psi^\star]^2 \kappa(\mu^s) \sum_{t=1}^{T} \sum_{x \in \mathcal{X}} \eta^t(x) \mathbb{I}_{\left\{x = x_h^t\right\}} \frac{1}{\mu_{1:}^s(x)} \kappa(\mu^s|x)$$

$$\leq [(1 + \lambda_1) C_\Psi C_\Psi^\star \kappa(\mu^s)]^2 \sum_{t=1}^{T} \sum_{x \in \mathcal{X}} \eta^t(x) \mathbb{I}_{\left\{x = x_h^t\right\}}$$

$$\leq [(1 + \lambda_1) C_\Psi C_\Psi^\star \kappa(\mu^s)]^2 \lambda_2 |\mathcal{X}| \sqrt{T} .$$

*Conclusion:* By summing these two upper bounds we get

$$\hat{\mathfrak{R}}^T \leq \left[ C_\Psi \lambda_2 \kappa(\mu^s) + [(1 + \lambda_1) C_\Psi C_\Psi^\star \kappa(\mu^s)]^2 \lambda_2 |\mathcal{X}| \right] \sqrt{T}$$

$$\leq 2 [(1 + \lambda_1) C_\Psi C_\Psi^\star \kappa(\mu^s)]^2 \lambda_2 |\mathcal{X}| \sqrt{T} .$$

The bound is finally obtained using the Theorem 2.2 that holds with a probability of at least $1 - \delta$ and links the estimated regret to the true regret.

$\square$

