# OpenReview forum: "Local and Adaptive Mirror Descents in Extensive-Form Games"
_NeurIPS.cc/2024/Conference — NeurIPS 2024 poster_

### Official Review · Reviewer_HP7A · 2024-06-23

**Soundness:** 3
**Presentation:** 3
**Contribution:** 2
**Rating:** 7
**Confidence:** 3

**Summary:**

The submission considers the problem of learning epsilon-Nash equilibria from trajectory feedback in zero-sum extensive-form games. The submission focuses on developing that avoids importance sampling over action sequences.

**Strengths:**

The submission is well-written, and the problem and solution are reasonable. The experiments are reasonable choices to demonstrate the efficacy of the solution.

**Weaknesses:**

My main criticism of the submission is that it purports its motivation to be motivated by solving large games. However, using a fixed sampling is untenable for learning good policies in large games, as the target policy and the behavioral policy will become too far apart. Model-free approaches to learning equilibria in large games (i.e., [1, 2]) learn on policy. It would be good for the submission to discuss this discrepancy and how it can be resolved in greater detail. I'll mention that the submission does already acknowledge this point and include some discussion at the end.

Also, it is worth noting that [2] showed empirical convergence for an on-policy algorithm with trajectory feedback and no importance sampling over action sequences. It would be interesting for the submission to discuss the feasibility of providing theoretical grounding for those results.

[1] From Poincaré Recurrence to Convergence in Imperfect Information Games: Finding Equilibrium via Regularization (2020)

[2] A Unified Approach to Reinforcement Learning, Quantal Response Equilibria, and Two-Player Zero-Sum Games (2023)

One other comment I had:

> Existing procedures suffer from high variance due to the use of importance sampling over sequences of actions (Steinberger et al., 2020; McAleer et al., 2022).

This claim and citation is a bit confusing. The whole point of McAleer et al. (2022) was to address this issue. The submission is citing a paper that refutes the claim of the sentence.

**Questions:**

I guess I'd be curious to hear the author's thoughts on whether providing grounding for [2]'s results is a promising direction.

**Limitations:**

I'll mention a limitation of my own review here: I did not confirm the correctness of the theory, so I'll defer to the other reviewer's on that point.

---

> ### Author Rebuttal · Authors · 2024-08-06
>
> We thank Reviewer HP7A for the overall positive review and the interesting references. We would like to answer the remarks and the question below.
>
> > My main criticism of the submission is that it purports its motivation to be motivated by solving large games. However, using a fixed sampling is untenable for learning good policies in large games, as the target policy and the behavioral policy will become too far apart. Model-free approaches to learning equilibria in large games (i.e., [1, 2]) learn on policy. It would be good for the submission to discuss this discrepancy and how it can be resolved in greater detail. I'll mention that the submission does already acknowledge this point and include some discussion at the end.
>
> Indeed, similarly to [3], we do not expend too much on this point as the best choices may not be motivated by theory. While the balanced policy gives the best rates, it may not be the optimal choice in practice. As we played a bit with the sampling policy in our experiments, we realized that updating from time to time the sampling policy with the current average one also provides good empirical results. We conjecture that for practical applications, the choice would not matter that much, as long as it decently explores the tree.
>
> This is why we mention the idea of restarting and taking the average policy as the new sampling one, as it achieves this purpose. We did not want to delve too much into this matter, as we think that this question would be better answered by more practical works, but we propose to add these insights to the paper.
>
> > Also, it is worth noting that [2] showed empirical convergence for an on-policy algorithm with trajectory feedback and no importance sampling over action sequences. It would be interesting for the submission to discuss the feasibility of providing theoretical grounding for those results.
>
> We thank Reviewer HP7A for pointing out these empirical results and we will add this reference in our literature review.
> Before answering, we would like to point out some differences between their theoretical setting and ours. First, they have a strong monotonicity assumption for the underlying operator $G$ they consider (in our case, $G$ would be $G:(\mu_{1:},\nu_{1:})-> (-\ell^\nu,\ell^\mu)$). This strong monotonicity assumption does not hold for their experiments, as the operator $G$ is linear with respect to the realization plan, hence they need to add some regularization, parameterized by the $\alpha$ constant. Second, they consider a full feedback setting in theory, observing the outcomes of all possible trajectories at each episode.
>
> Despite these differences, we found out that their approach can still be adapted to our setting. An entropic regularization with the Kullback-Leibler divergence allows for the use of an importance sampling estimator, which gives an unbiased estimator of $G$ (formerly provided by the full feedback). The rate is no longer linear, but only in $\mathcal{O}(T^{-1/2})$ for the exploitability with a fixed regularization. Then, as mentioned in [2] in commentary of Figure 2, the regularization can be decreased over time to converge to the real problem, but we think only a rate of $\mathcal{O}(T^{-1/4})$ for the true exploitability can be achieved with this approach, by taking $\alpha$ also of order $\mathcal{O}(T^{-1/4})$. However, this approach again introduces the importance sampling over action sequences to deal with the trajectory feedback.
>
> If the regularization is taken such that it compensates for the importance sampling term, then we obtain their empirical approach for dealing with the trajectory feedback. This still implies the need for an analysis with a regularization that varies with time, as the current policies will change. We think that the fact the analysis is done iteratively (as in their Theorem 3.4) rather than being based on the regrets helps in this regard, but we do not know yet if it is possible to provide some theoretical guarantees for this approach.
>
>
> >One other comment I had: "Existing procedures suffer from high variance due to the use of importance sampling over sequences of actions (Steinberger et al., 2020; McAleer et al., 2022)." This claim and citation is a bit confusing. The whole point of McAleer et al. (2022) was to address this issue. The submission is citing a paper that refutes the claim of the sentence.
>
> We cited McAleer et al. (2022) as it raises the issue and tries to solve it using the fixed sampling framework considered in our submission but with a CFR-based algorithm. As explained later in the introduction, existing procedures *not relying on fixed sampling* suffer from the high variance issue. We acknowledge this sentence could be a bit confusing and we will replace it by:
>
> "As noted by McAleer et al., 2022 most of the existing procedures suffer from high variance due to the use of importance sampling over sequences of actions.”
>
> [3] McAleer, S., Farina, G., Lanctot, M., and Sandholm, T. ESCHER: Eschewing importance sampling in games by computing a history value function to estimate regret, International Conference on Learning Representations 2022

---

> > ### Comment · Reviewer_HP7A · 2024-08-09
> > **Response**
> >
> > > We thank Reviewer HP7A for pointing out these empirical results and we will add this reference in our literature review. Before answering, we would like to point out some differences between their theoretical setting and ours. First, they have a strong monotonicity assumption for the underlying operator they consider (in our case, would be). This strong monotonicity assumption does not hold for their experiments, as the operator is linear with respect to the realization plan, hence they need to add some regularization, parameterized by the constant. Second, they consider a full feedback setting in theory, observing the outcomes of all possible trajectories at each episode.
> >
> > There may be a misunderstanding about the point I was making here. I was specifically pointing to their experiments with black box feedback (not full feedback) in Figure 13. The reference provides no theory for these experiments, but I think they are still interesting because they seem to be converging 1) on policy, 2) without importance sampling over sequences, 3) using black box feedback. I would be curious to hear any speculation the authors' may have on whether or not deriving guarantees for algorithms having those properties is a promising direction for this literature.

---

> > > ### Author Response · Authors · 2024-08-13
> > >
> > > Sorry, our response to this remark may not have been very clear. The first two paragraphs were about their theoretical setting (with a regularized full feedback and on-policy), but the third paragraph was indeed referring to these experiments with trajectory feedback, trying to bridge the gap between the two.
> > >
> > > We think that proving guarantees for their algorithm would be a very good contribution, but we do not know yet if it is doable (maybe this would not work well on carefully engineered games).
> > >
> > > The main difficulty is the regularization: this kind of update can make sense theoretically if the regularization is chosen to compensate for it at each iteration. However, this also implies that the solution of the regularized game changes over time, which is hard to analyze.

---

### Official Review · Reviewer_yah9 · 2024-07-13

**Soundness:** 3
**Presentation:** 3
**Contribution:** 3
**Rating:** 7
**Confidence:** 2

**Summary:**

The paper studies the extensive-form game under the fixed sampling policy framework. It proposes the algorithm based on online mirror descent. The paper gives near-optimal regret bounds for the proposed algorithm, under different learning rate settings. The algorithm is justified with experiments.

**Strengths:**

1. The the paper has detailed introduction and literature review, with a clear description of the problem setting and framework. The paper also makes sufficient comparisons to existing works under the same or different framework.

2. The result has optimal dependency on $T$ and near-optimal dependency on other game-related parameters. The removal of the importance sampling term in the fixed-rate setting is a good contribution.

3. The paper justifies it's theory with experiments. It makes comparison with a benchmark method and experiment result looks convincing.

**Weaknesses:**

1. The framework itself is still confusing. While the authors made some discussion of the framework in section 2.2, the advantage of fixed sampling policy seems not reflected in the paper's theoretical result. And in terms of the regret, the paper's result doesn't improve the regret of simultaneous regret minimization procedures.

2. On the explanation of the theory, the paper can be improved by adding comparison to more benchmarks. The paper only compares the result with the result of simultaneous regret minimization procedure, and the conclusion is the bound matches but doesn't improve the existing result. It could be helpful to understand the paper's result by comparing previous results under the same fixed sampling policy framework.

**Questions:**

From the theorems the fixed and adaptive rate bounds have the same dependency on $T$. Is it correct to understand the motivation of the adaptive rate is only to remove the dependency on the importance sampling term $\kappa$? When the balanced policy is used as the sampling policy, $\kappa$ is reduced to a game-dependent term. In this case, is it correct think the adaptive rate setting as unnecessary?

**Limitations:**

/

---

> ### Author Rebuttal · Authors · 2024-08-06
>
> We thank Reviewer yah9 for the overall positive review. We would like to answer the points made below.
>
> > The framework itself is still confusing. While the authors made some discussion of the framework in section 2.2, the advantage of fixed sampling policy seems not reflected in the paper's theoretical result. And in terms of the regret, the paper's result doesn't improve the regret of simultaneous regret minimization procedures.
>
> The optimal rate in the online case is $\tilde{\mathcal{O}}(H(A_X+B_Y)/\epsilon^2))$ and is attained up to constant and logarithmic factors in [1] (note that the proof of the lower bound also works in the fixed sampling framework). For this reason, we cannot hope for theoretical improvements in this setting.
>
> The purpose of fixed sampling is to remove the importance sampling term, that does not appear in the leading theoretical regret term but that would make (in practice) function approximations fail. Removing this problematic term requires different ideas and analysis.
>
> > On the explanation of the theory, the paper can be improved by adding comparison to more benchmarks. The paper only compares the result with the result of simultaneous regret minimization procedure, and the conclusion is the bound matches but doesn't improve the existing result. It could be helpful to understand the paper's result by comparing previous results under the same fixed sampling policy framework.
>
> We are not aware of other papers that work within the fixed sampling framework with trajectory feedback besides those we have mentioned. If Reviewer yah9 can point additional references, we would be pleased to include them. Consequently, we primarily compare our theoretical results with those of [2]. Reference [3] only achieves a convergence rate of $\mathcal{O}(1/\epsilon^2)$ with respect to $\epsilon$. In the introduction, we discuss the optimal rate already achieved in the online setting by [1].
>
> We will make an effort to reference these results more explicitly in our discussion.
>
>
> > From the theorems the fixed and adaptive rate bounds have the same dependency on $T$
> . Is it correct to understand the motivation of the adaptive rate is only to remove the dependency on the importance sampling term
> ? When the balanced policy is used as the sampling policy,
> $\kappa$ is reduced to a game-dependent term. In this case, is it correct think the adaptive rate setting as unnecessary?
>
> The motivation behind the adaptive rate, in addition to being a more natural choice, is mostly practical: it does not require the balanced policy to be computed beforehand and the horizon $T$ to be known. Theoretically, the adaptive rate is indeed unnecessary, as using the balanced policy already allows for the best rate with respect to the game parameters. Nevertheless, considering the practical advantages of an adaptive policy, we wanted to also provide some theoretical guarantees for it.
>
> Our initial thought was that we could obtain the same optimal rate. However, we did not manage to prove it and now strongly believe it is not possible. Optimality with regards to the $\epsilon$ dependency was possible nonetheless under some quite general assumptions (with fewer restrictions on the fixed sampling policy and the regularizer) which is already in our opinion a valuable result.
>
> [1] Fiegel, C., Ménard, P., Kozuno, T., Munos, R., Perchet, V., and Valko, M. Adapting to game trees in zero-sum imperfect information games, International Conference on Machine Learning, 2023
>
> [2] Bai, Y., Jin, C., Mei, S., and Yu, T. Near-optimal learning of extensive-form games with imperfect information, International Conference on Machine Learning, 2022
>
> [3] McAleer, S., Farina, G., Lanctot, M., and Sandholm, T. ESCHER: Eschewing importance sampling in games by computing a history value function to estimate regret, International Conference on Learning Representations 2022

---

### Official Review · Reviewer_1VRY · 2024-07-13

**Soundness:** 3
**Presentation:** 2
**Contribution:** 3
**Rating:** 6
**Confidence:** 4

**Summary:**

I reviewed an earlier version of this paper. While I have some additional comments, my review is largely similar to my previous review, since the paper has only a few minor edits relative to the previous version, as far as I can ascertain.

The paper introduces algorithms designed to approximate Nash equilibria in zero-sum imperfect information games (IIG) with trajectory feedback. Specifically, the authors focus on the fixed-sampling framework. Past approaches have struggled with significant variance, largely due to the utilization of importance sampling across action sequences (McAleer et al. (2022) provide an algorithm that doesn't require importance sampling, but their techniques don't generalize to non-scale invariant regret-minimization-based algorithms such as OMD).

The proposed approach employs an adaptive Online Mirror Descent (OMD) algorithm. This involves using adaptive learning rates along with a dilated regularizer. The paper demonstrates that this method ensures a convergence rate of O\tilde(T^{-\frac{1}{2}) with high probability and exhibits a near-optimal dependence on the game parameters, when employed with appropriate choices of sampling policies.

**Strengths:**

The paper provides a strong technical contribution in producing an adaptive trajectory-feedback-based adaptive OMD algorithm that doesn't rely on importance sampling and generalizing DS-OMD to use time-varying regularization.

The paper provides empirical evidence for the convergence and variance of their approach, compared to other approaches in the literature, and also provides code for their algorithm.

**Weaknesses:**

The paper could delineate its contributions better. As noted by most reviewers in a previous submission, the similarity to the regret circuit decomposition of CFR is apparent (and noted by the authors). Still, the difference could be further highlighted in the contributions, especially since this was confusing for several reviewers last time. A note was made by the authors last time regarding the interpretation of their method as regularization at the global level (whereas CFR doesn't have this sort of interpretation). While it is mentioned now that the proposed algorithm is an instance of GDS-OMD, I think these things could be further emphasized throughout the paper.

Along the same lines, one of the reviewers pointed out last time that things like $h_t$ and $q_t$ are not defined in the main body, and as far as I can tell, this is still an issue, even though the authors indicated that this would be clarified (even if not at the detailed technical level as done in the appendix) in the main body. It would be good for the authors to implement these changes, given that they assured the reviewers they would do this to improve the presentation.

One of the reviewers noted last time that the motivation for variance reduction is unclear. I agree with this. While the reviewers mention that variance reduction becomes a concern for performance in function approximation settings, it would be nice to see this in their experiments (since the current empirical evidence for the provided algorithm doesn't seem to indicate that there is much gain in performance beyond the variance reduction with the current approach).

**Questions:**

No specific questions beyond the weaknesses mentioned above.

**Limitations:**

Yes

---

> ### Author Rebuttal · Authors · 2024-08-06
>
> We first thank Reviewer 1VRY for taking the time to review our submission and for suggesting improvements. We address them below.
>
> > The paper could delineate its contributions better. As noted by most reviewers in a previous submission, the similarity to the regret circuit decomposition of CFR is apparent (and noted by the authors). Still, the difference could be further highlighted in the contributions, especially since this was confusing for several reviewers last time. A note was made by the authors last time regarding the interpretation of their method as regularization at the global level (whereas CFR doesn't have this sort of interpretation). While it is mentioned now that the proposed algorithm is an instance of GDS-OMD, I think these things could be further emphasized throughout the paper.
>
> LocalOMD indeed enjoys at the same time a local interpretation (as CFR) and a global interpretation (as the application of GDS-OMD updates). This is actually a unique feature of our algorithm.
> As noted by Reviewer 1VRY, we mention this particularity several times:
> - At the beginning of the contribution section (l. 81), pointing the similarity to a paper studying a CFR approach.
> - In the local loss paragraph (l. 251), dedicated to this idea.
> - For justifying the use of the adaptive rates (l. 273).
>
> We propose to add to the conclusion the following sentence:
>
> “LocalOMD enjoys simultaneously two interpretations: one as a Mirror Descent type algorithm working at the global level, with a single update performed at each iteration over the whole tree; and one as regret minimizers working locally at each information set, which makes it very similar to a CFR algorithm despite a fundamentally different approach.”
>
> > Along the same lines, one of the reviewers pointed out last time that things like
> $h_t$ and $q_t$ are not defined in the main body, and as far as I can tell, this is still an issue, even though the authors indicated that this would be clarified (even if not at the detailed technical level as done in the appendix) in the main body. It would be good for the authors to implement these changes, given that they assured the reviewers they would do this to improve the presentation.
>
> Indeed, the notations $h_t$ and $q_t$ are only defined in Algorithm 2 in the main paper, and not directly in the body. We tried to include them in the main text after the last submission with some necessary explanations of where these two terms come from to avoid further confusion. In particular we would need to reproduce appendix D.1 of [1] or appendix F of [2] about obtaining a practical implementation of the GDS-OMD update. However, this would obfuscate the main message of this paper with technical details already appearing in the literature. That is why we finally decided to skip them.
> If a reviewer thinks that those explanations should be inserted here from existing papers, we could use the extra-page to include them nonetheless.
>
>
> > One of the reviewers noted last time that the motivation for variance reduction is unclear. I agree with this. While the reviewers mention that variance reduction becomes a concern for performance in function approximation settings, it would be nice to see this in their experiments (since the current empirical evidence for the provided algorithm doesn't seem to indicate that there is much gain in performance beyond the variance reduction with the current approach).
>
> The high scale of the rewards is an issue that currently prevents the practical implementation of Mirror Descent algorithm with trajectory feedback: as explained at l. 258, the reward can be of order $\mathcal{O}(A_X)$, which is incompatible with the function approximation settings. The main argument favoring the fixed sampling framework is that it allows the rewards to stay of order $\mathcal{O}(HA)$, which becomes manageable.
>
> We have included experiments demonstrating that this approach does indeed reduce variance in relatively small games. However, this issue of high variance is less significant in the tabular setting. Additionally, demonstrating that this modification enables function approximation with the trajectory-based Mirror Descent approach would require a separate paper, given the extensive technical adjustments needed to adapt a tabular algorithm to the function approximation setting, which is beyond the scope of our current submission.
>
>
> [1] Bai, Y., Jin, C., Mei, S., and Yu, T. Near-optimal learning of extensive-form games with imperfect information, International Conference on Machine Learning, 2022
>
> [2] Fiegel, C., Ménard, P., Kozuno, T., Munos, R., Perchet, V., and Valko, M. Adapting to game trees in zero-sum imperfect information games, International Conference on Machine Learning, 2023

---

> > ### Comment · Reviewer_1VRY · 2024-08-13
> >
> > Thank you for your detailed response; my understanding of your paper and contribution has improved and I have raised my score.
> >
> > I hope that the authors consider the suggestions provided by reviewers from past submissions and the current submission. Going back to suggestions pointed out in a review of an earlier submission, even the very minor point of correcting the stylization of the game "Liar's Dice" was not corrected, even though this was acknowledged by the authors as something to be fixed. I only bring this up because as I noted in my review there was not much change in this submission relative to the last submission that I reviewed (indeed, this is why I pointed out "weaknesses" corresponding to discussion between past reviewers and the authors); I hope that if the paper gets accepted, the authors end up implementing the changes they suggest they will in their rebuttals.
> >
> > > “LocalOMD enjoys simultaneously two interpretations: one as a Mirror Descent type algorithm working at the global level, with a single update performed at each iteration over the whole tree; and one as regret minimizers working locally at each information set, which makes it very similar to a CFR algorithm despite a fundamentally different approach.”
> >
> > I agree that would make a good addition to the conclusion.
> >
> > > That is why we finally decided to skip them. If a reviewer thinks that those explanations should be inserted here from existing papers, we could use the extra-page to include them nonetheless.
> >
> > Makes sense. I agree that it is not necessary to include them in the main body to prevent the obfuscation of the contribution of this paper.

---

### Decision · Program_Chairs · 2024-09-25

**Decision:**

Accept (poster)

**Comment:**

The paper introduces an algorithm to approximate Nash equilibria in zero-sum imperfect information games (IIG) with trajectory feedback using the fixed-sampling framework. They obtain a high-probability convergence rate.
The reviewers unanimously vote for accepting the paper and are quite enthusiastic about the current submission.